# A Dark Target research aerosol algorithm for MODIS observations over eastern China: Increasing coverage while maintaining accuracy at high aerosol loading

Yingxi R. Shi[1,4], Robert C. Levy[1], Leiku Yang[3], Lorraine A. Remer[4], Shana Mattoo[1,2], and Oleg Dubovik[5]

[1]NASA Goddard Space Flight Center, Greenbelt, MD, USA
[2]SSAI, Lanham, MD, USA
[3] School of Surveying and Land Information Engineering, Henan Polytechnic University, Jiaozuo 454003, China
[4]UMBC/JCET, Baltimore, MD, USA
[5]Univeristy of Lille, Lille, France

*Correspondence to*: Yingxi Shi (yingxi.shi@nasa.gov); Leiku Yang (yanglk@hpu.edu.cn)

**Abstract.** Satellite aerosol products such as the Dark Target (DT) produced from the MODerate resolution Imaging Spectroradiometer (MODIS), are useful for monitoring the progress of air pollution. Unfortunately, the DT often fails to retrieve during the heaviest aerosol events as well as the more moderate events in winter. Some literatures attribute this lack of retrieval to cloud mask. However, we found this lack of retrieval is mainly traced to thresholds used for masking of inland water and snow. Modifications to these two masks greatly increase the coverage of retrievals overall (50%) and double the retrievals of aerosol optical depth at 0.55 µm (AOD) greater than 1.0. The "extra" high AOD retrievals tend to be biased when compared with ground-based sunphotometer (AERONET). Reducing bias in new retrievals requires two additional steps. One is an update to the assumed aerosol optical properties (aerosol model) – the haze in this region is both less absorbing and lower in altitude than what is assumed in the global algorithm. The second is accounting for the scale height of the aerosol, specifically that the heavy aerosol events in the region are much closer to the surface than what is assumed by the global DT algorithm. The resulting combination of modified masking thresholds, new aerosol model, and lower aerosol layer scale height was applied to three months of MODIS observations (Jan-March 2013) over eastern China. When compared with AERONET, 70% of the research algorithm retrievals fall within ±(0.08+0.17×AOD). We also find that the research algorithm is able to identify additional pollution events that a triad of AERONET instruments surrounding Beijing could not. Mean AOD retrieved from the research algorithm increases from 0.11 to 0.18 compared to values calculated from the operational DT algorithm during January to March of 2013 over the study area. But near Beijing where the severe pollution occurs, the new algorithm increases AOD by as much as 3.0 for each 0.5° grid box, over the previous operational algorithm values.

**1 Introduction**

Because of rapid economic development and urbanization, eastern China, especially surrounding the Beijing area, has a large aerosol emission and complex aerosol composition. The resulting large aerosol loading creates serious air quality and public health problems (Zhang et al., 2012, Huang et al., 2014). Air quality issues in this region produce significant social economic impact due to the high density and continuous increase of the population and energy consumption. For example, Beijing alone had 12 million residents in 1998, with the number increasing rapidly to 21.51 million by the end of 2014 (Beijing Municipal Statistics Bureau). Studies show that the annual number of "haze days" over the North China Plain, defined as a day with a visibility < 3 km (QX/T 113-2010), increased from 1980 and steeply increased since 2001 (Chen et al., 2015, Ding et al., 2014). In January 2013, eastern China experienced their worst ever severe haze/fog event. This event was marked by extremely high level of PM2.5 (particulate matter with an aerodynamic diameter less than 2.5 μm), long duration (70% of the days in January exceeding the Chinese air quality standard of PM2.5 = 75 μg/m$^3$), extended spatial coverage (~1.3 million km$^2$), which affected ~800 million people (Renhe., 2014, Huang et al., 2014). During this episode, Beijing's hourly PM2.5 exceeded 600 μg/m$^3$, which is nearly 60 times higher than the World Health Organization's (WHO) 'good' standard (WHO, 2006).

To try and mitigate this severe pollution, the Chinese government launched an air pollution control program in 1998. More drastic measures were taken during the 2008 Beijing Olympics (Chen et al., 2013) and also after the severe haze events of January 2013. Specifically, after the 2013 event, the Chinese State Council released the 5-year Clean Air Action Plan aiming to reduce annual PM2.5 to less than 60 μg/m$^3$ by 2017 (Huang et al., 2014). To assess this achievement, the aerosol monitoring network was expanded quickly. While ground-based sites were added, there began a huge effort to utilize satellite aerosol products.

Even though satellite aerosol products represent a total-column optical measurement (aerosol optical depth – AOD) and not the PM2.5 measurements required for air quality concerns, the birds-eye vantage of the satellite has been shown to be useful. The Dark Target (DT) aerosol products on the Moderate Resolution Imaging Spectroradiometer (MODIS) sensors onboard both Terra and Aqua satellites has become a popular dataset used in studies of Chinese aerosol (Christopher and Zhang, 2002; Li et al., 2004; Yu et al., 2004; Quaas et al., 2008; Zhang et al., 2012; Costantino et al., 2013; Luo et al., 2014; Bellouin et al., 2020; ). However, the DT products miss aerosol retrievals over eastern China, especially during wintertime. Many studies have discussed this issue (Yan et al., 2016, Bilal et al., 2014, 2015, Wei et al., 2019). Some attributed the problem of lack of retrieval to an overly aggressive cloud mask, which is not what we found in this study. Most of these studies tried to solve the missing data problem by using extra ancillary data or by developing a new algorithm (Yan et al., 2016, Li et al., 2012, Wei et al., 2019). Some of these methods are only applicable over a very small region, such as near the Beijing-Tianjing-Hebei area (Bilal et al., 2015). None of these methods can provide real time AOD, consistent with the DT products. Near real time data is critical in terms of air quality forecasting and policy making, and the desire to be consistent

with the DT product ties local characterization to a global standard. Due to the importance of satellite aerosol products for air quality and aerosol forecasting, our goal is to provide a more comprehensive near real time DT AOD map over China. The main problem with the current DT product for China is lack of retrievals. Even when the scene should be acceptable for an aerosol retrieval, the current operational algorithm fails to return a result. We will show in this paper why there are so many missing retrievals, and we will present a regionally specific research algorithm that remedies the problem and fine tunes the algorithm for a more accurate product. The advantage of this regional research algorithm is that it is built within the structure of the DT algorithm so that implementation into the global operational system will be less onerous. However, implementation into the operational system is beyond the scope of this paper. In Section 2, we describe the data that will be used in this study, and in Section 3 we illustrate the problem with the current operational algorithm with two case studies. This is followed by a presentation of a new research algorithm for China that (a) increases retrieval coverage in the region and (b) makes other adjustments so that high accuracy can be maintained as coverage is increased (Section 4). The new research algorithm is validated in Section 5, and in Section 6 we use the new algorithm to characterize an extreme pollution event. Section 7 summarizes and presents the conclusions.

## 2 Description of Data Products

### 2.1 MODIS Dark Target aerosol products and algorithm

The MODIS Dark Target algorithm, based on a lookup table (LUT) method, uses three wavelengths from 0.47 to 2.1μm to retrieve aerosol properties over dark (vegetated and dark-soiled) land surfaces (Levy et al., 2007ab, 2013; Remer et al., 2020). The basics of the over-land algorithm are as follows: 1) consolidate higher spatial resolution (e.g. 500 meter and 1 km) calibrated reflectance and radiance observations (known as Level 1B 'pixels') into 10 km retrieval 'boxes'. 2) apply various filters to remove ('mask') clouds, cloud shadows, snow and ice, inland water, and any pixels that represent conditions that are not suitable for retrieval (Remer et al., 2005). 3) assign the aerosol model (optical and physical properties) that are most likely representative of a given season and location, 4) assume reflectance properties of the surface using an empirically derived equation, 5) search the pre-calculated LUTs simulating the observations of different loadings of the assumed aerosol type, 6) report the total aerosol loading (AOD) that combined with the surface reflectance, provides the best match with the observed spectral reflectance, and finally 7) assign quality assurance and other diagnostics to the retrieval product. In a global sense, the generalized aerosol retrieval, along with strict quality assurance (e.g. QA confidence values = 3) has been shown to provide accurate retrievals and reasonable coverage over most conditions. For global retrievals over land, Expected Errors tend to be on the order of $\Delta\tau = \pm(0.05 + 0.15\tau)$.

Although generally appropriate on a global scale, the thresholds for filtering/masking in step 2 can be too weak or too strong for particular regions. If too strong, they will falsely mask out legitimate aerosol retrievals. For land surfaces with vegetation, there is a strong absorption of radiation in the shortwave infrared (SWIR) by liquid water, and therefore the land surface parameterization is based on the assumption that liquid water in the leaves of vegetation is correlated with the pigments in

those leaves that absorb radiation in the visible (blue and red wavelengths) during photosynthesis. However, liquid water not in vegetative structures (such as inland water) will also absorb SWIR wavelengths, confuse the algorithm and therefore must be masked. While designed to monitor the health of green vegetation (Tucker, 1979), the Normalized Difference Vegetation Index (NDVI) can also be used to detect inland water. As defined as Eq. (1):

$$\mathrm{NDVI} = {\rho_{0.87}\text{-}\rho_{0.66}}/{\rho_{0.87} + \rho_{0.66}} \quad (1).$$

where ρ is the reflectance at TOA at 0.87 (NIR) and 0.66 μm (Red) bands.

The aerosol algorithm uses NDVI in an inverse fashion to exclude nonvegetative scenes that might have a thin layer of water on the surface, such as melting snow or swamps. Scenes with near very low values of NDVI (say < 0.1) include inland water, cloud edges, and arid regions such as deserts. Therefore, NDVI is an overall powerful tool that masks out many other conditions that are not optimal for applying the DT algorithm. The problem is that by increasing reflectance more in the red band than in the NIR band, heavy loadings of fine-dominated aerosol types (as are found in eastern China) can also depress the values of NDVI (Yang et al., 2020).

Analogous to the NDVI, we can define a difference index for detecting snow. A Normalized Difference Snow Index (NDSI) is similar to the NDVI used for the inland water mask, but as described in Li et al. (2005) it is based on different wavelengths, given as Eq. (2)

$$\mathrm{NDSI} = {\rho_{0.87}\text{-}\rho_{1.24}}/{\rho_{0.87} + \rho_{1.24}} \quad (2).$$

where ρ is the reflectance at TOA at 0.87 (NIR) and 1.24 μm. This NDSI relies on the strong absorption and reflectance features of snow and ice, and in tandem with a brightness temperature threshold (e.g. 11 μm channel), is used to mask out snow/ice. The operational DT algorithm considers pixels with NDSI > 0.01 and 11 μm channel brightness temperature less than 285K to be snow, melting snow, or contaminated pixels near snow edges. Note that the temperature threshold is above the freezing level, which is a cautious approach to include regions where snow is in the process of melting and may be above freezing.

Overall, both the inland water (NDVI) and snow/ice (NDSI + temperature) filters are designed to optimize the balance between accuracy and availability of aerosol retrievals, on a global basis, but may not be ideal for a specific region such as China, where light to heavy pollution conditions occur year-round and particularly during the winter season.

The LUT used in the DT retrieval algorithm is calculated from prescribed aerosol models. The prescribed models depend on season and region, and are based on global categorizing of Aerosol Robotic Network (AERONET) inversion products (Dubovik and King, 2000) as described in Levy et al., (2007). In operation, the aerosol model assumed for most of China and for most seasons is the 'moderately absorbing' model having single scattering albedo, $\omega_0$, at 0.55 μm, around 0.90. During the summer and fall along China's coastal belt, as for a small region around Taiwan in all seasons, the 'weakly-absorbing' ($\omega_0 \sim 0.95$) model is used instead. These models were assigned to China for the version of the algorithm that went into production in 2005 (Collection 5), and while re-evaluated for Collection 6 that began in 2013, they were not adjusted.

However, with the continued development of China's industry and urbanization as well as their "clean air" movement after 2008, the aerosol composition may have changed significantly enough to warrant an update of aerosol model selection.

In addition to the aerosol models being prescribed, the LUTs are calculated assuming a vertical profile for the aerosol. Except for coarse dust aerosol, all fine-dominated aerosol types (including the moderate and weakly absorbing types used for China), are assumed to have a scale height (H) of 2.0 km. In fact, pollution aerosol in China, especially during the winter months, tends to form under extremely stable conditions. Studies (Tang et al., 2015, Li et al., 2015, Luan et al., 2018; Liu et al., 2015) indicate that scale heights for the haze can be significantly less than 1.0 km, which due to multiply scattering interactions with the molecular atmosphere, lead to errors in the LUTs assumed to simulate satellite observations.

The combination of less than ideal filtering for inland water and snow/ice, of changing aerosol composition in the last two decades, and wrong assumptions of aerosol scale height, lead to systematic errors in both coverage and accuracy over China.

### 2.2 Other MODIS aerosol products

DT is not the only algorithm that makes use of MODIS observations to derive aerosol properties. There are two additional AOD algorithms, known as "Deep Blue" (DB) and "Multi-Angle Implementation of Atmospheric Correction" (MAIAC) Since each algorithm uses different criteria for filtering and masking, makes different assumptions regarding aerosol optical properties and surface reflectance, and uses different techniques for fitting spectral observations, we can examine them and their products to inform possible solutions to the China retrievals outside of AERONET's coverage.

The DB algorithm, as its name indicates, uses observations in the "Deep Blue" or near-ultraviolet (NUV) part of the spectrum (~0.412 μm) in addition to observations in the visible blue (0.466 μm) and red wavelengths (Hsu et al., 2006). DB bands can capture the aerosol signals due to that carbonaceous aerosol types have strong absorption in shorter wavelengths and desert surfaces have weaker reflectance in these wavelengths. For MODIS, the DB product has the same spatial resolution (10 x 10 km) as the DT product, and has reported uncertainties (for highest quality assurance) defined by Eq. (3):

$$\Delta\tau = \pm([0.086 + 0.56\tau_{DB}]/[1/\mu_0 + 1/\mu]) \quad (3)$$

where $\mu_0$ and $\mu$ are the cosine of the solar and view zenith angles, respectively (Sayer et al., 2013). For heavy smoke including pollution if very optically dense, DB developed a smoke detection scheme based on Lambertian equivalent reflectivity (Dave and Mateer, 1967) at 0.412 μm, 0.488 μm and 0.672 μm, as well as brightness temperature at 11 μm. Once the aerosol is classified as smoke, the cloud mask is relaxed to ensure good retrieval spatial coverage and the spatial variability threshold is also relaxed when assigned the data quality (Hsu et al., 2019).

The MAIAC algorithm, uses time series (up to 16 days) analysis to exploit multangular information of atmosphere and land surface in order to derive semi-empirical bidirectional reflectance functions of surface (Lyapustin et al., 2014, 2018). The algorithm utilizes the fact that the surface is more static than the atmosphere components such as clouds or aerosols during a short time span. With a more accurate description of surface characteristics, MAIAC has better ability to retrieve very optically thick aerosol plumes (Mhawish et al., 2019). However, data gaps still exist in cloud free region due to terrain or high surface albedo issues (Bi et al, 2019). MAIAC produces a product with the much finer spatial resolution of 1 km and

has a reported bulk uncertainty of 66% of retrievals within $\pm 0.05 \pm 0.1AOD$ (Lyapustin et al., 2018). The MAIAC atmospheric product (MCD19A2) is not stored in a traditional format of MODIS granule (e.g. retrievals along the native swath), but uses the MODIS Sinusoidal grid instead (Stackpole, 1994). All granules are regraded into a 1 km sinusoidal map and the overpass time, based on the granule ID, is stored in the product. Although there are slight differences between the time stamp and the granule ID (personal communication with Dr. Yujie Wang), which becomes apparent when comparing the three MODIS aerosol products in case studies. The differences will not statistically affect the comparison in our region of interest.

### 2.3 AERONET sun and sky aerosol products

The AErosol RObotic NETwork (AERONET) is a global aerosol-monitoring network that is commonly used as a benchmark for validating satellite-retrieved AOD and to study the aerosol properties globally (Holben et al., 1998; Levy et al., 2013; Remer et al., 2005; Sayer et al., 2013; Zhang and Reid, 2006, Shi et al., 2011, 2013; Giles et al., 2019). AERONET provides two aerosol products. One measures aerosol attenuation through direct sun measurements, which provides spectral aerosol optical depth every 3 or 15 minutes (Holben et al., 1998). The most current version of this product is the version 3 product (Giles et al., 2019), which changes the cloud screening procedures from the older version and includes more AOD observations that are higher than 1.0 (Eck et al., 2018; Eck et al., 2019). This change is critical to evaluate satellite performance over regions with high AOD loading. The new version product also has better cirrus filters and more accurate quality control procedures. The uncertainty in AOD from version 3 remains the same as previous versions, which is ~0.01 in the visible and near-infrared and ~0.02 at ultraviolet (UV) wavelengths (Eck et al., 1999).

In addition, AERONET instruments measure sky radiance and provide inversion products that contain aerosol microphysical and optical properties, such as particle size distribution, complex refractive index, and phase function (Dubovik and King, 2000; Dubovik et al., 2002; 2006). The new version 3 inversion products contain both traditional almucantar mode sky measurements and the new hybrid mode sky measurements. The almucantar mode is a series of measurements of the sky with changing azimuthal angles from $0^{o}$ to $\pm$ $180^{o}$ and a fixed solar elevation angle (Holben et al., 1998). Almucantar mode can only measure aerosol properties when the solar zenith angle (SZA: the complement of the solar elevation angle) is greater than 50°. That is when there is a sufficient range of scattering angles for high retrieval accuracy (Holben et al., 2006). This angle limitation is associated with the aerosol diurnal cycle, which means that there will be no aerosol properties derived during the middle of the day. The hybrid scan, which changes in both azimuthal and zenith angle directions simultaneously, can provide robust retrievals for measurements up to 25° SZA (Sinyuk et al., 2020). Because the hybrid scan is only available with the new CIMEL Model-T, the availability of hybrid measurements is limited, as compared with almucantar measurements. In our study, there are three sites that provide the hybrid scan: Beijing-CAMS, Beijing_PKU, and Yanqihu.

During this process, we used all available version 3 level 2 AOD and inversion products over China following QA procedures and recommendations found in Holben et al., (2006) to acquire a new aerosol model for the Beijing region as well as using AERONET data for validation purposes.

## 3. Case studies of high and low aerosol loading scenarios over the Beijing area

While the DT algorithm has been a proven success as a global product, there continue to be regions where the algorithm under performs, and one of those regions is China. Many studies find that the algorithm frequently fails to retrieve in situations. But we find there is no physical reason to prevent retrieval. These situations occur both in high and low AOD situations. We present two case studies to illustrate the problem.

### 3.1 An intense high AOD pollution event on October 9$^{th}$ 2013

Figure 1 illustrates a heavily polluted condition over East Asia, which is common over this region. Figure 1a is the MODIS 'true-color' reflectance image from 25° to 45° latitude and 105° to 145° longitude. Significant pollution aerosol plumes, appearing gray, cover Beijing (latitude 39.9ºN longitude 116.4º E) and its surrounding regions and extend towards the southwest all the way to the edge of the image. Clear patterns of variation in pollution can be found within the plume. Figure 1b is the MODIS DT AOD for all available retrievals, including retrievals meant for only qualitative imagery (QA = 0 to 3). The AOD gradually increases from very low loading outside of the plume to close to 1.0 at the edge of the pollution. However, there is no AOD retrieved at the thickest part of the plume. Two nearby AERONET sites, Xianghe and Beijing reported AOD at 0.5 µm from 2 to 3 and above 3, respectively. Figure 1c shows the MODIS DB AOD for all available DB retrievals (QA = 0 to 3). DB has retrieved similar AOD as compared with the DT product but has filled the DT data gap with mostly AOD of 3.0. This is the upper boundary for AOD in the DB retrieval as the AOD within the thickest part of the plume has very limited dynamic range and does not reflect the variation of AOD that is shown in Figure 1a. Figure 1d shows the all available MODIS MAIAC AOD. MAIAC product shows 1km resolution AOD and the majority of the region reports AOD around 1.5 to 2.0, which is lower than what DB reports. Over two small regions, AOD reaches 3.0 and above. We also checked the OMI UV AI, which showed that the pollution plume was not very absorbing in the UV part of the spectrum. Figure 1 demonstrates that all of the MODIS retrieval algorithms have trouble detecting and/or retrieving the heavy pollution. However, they appear to fail for different reasons.

A natural hypothesis is that the aerosol retrieval fails because it confuses a heavy aerosol plume with a cloud like many studies suggested (Mhawish et al., 2019, Bi et al., 2019, Tao et al., 2015, Yan et al., 2016, ). The DT algorithm, however, provides both a cloud fraction estimate as well as a quality assurance cascade to help determine the point in which the retrieval fails. Near the center of the plume where the pollution is heaviest (Figure 1d), the cloud fraction in the MODIS DT aerosol product is almost 0, which indicates that an overly aggressive cloud masking is not the major reason these aerosols are not retrieved. Instead, it is the NDVI map (Figure 1e) which explains why the retrieval fails. Here the NDVI values are less than 0.1 which used for inland water mask, denoted by the dark blue to purple at where the thickest pollution occurs.

These values are below the threshold of the inland water mask, which triggers the mask in operational procedures and prohibits retrieval there. This case study illustrates the problem of not being able to retrieve AOD over high optical depth pollution scenarios and its impact.

### 3.2 A low AOD pollution event on December 13th 2018

In addition to not retrieving high aerosol loading, it is also common for DT to miss moderate to low AOD cases during winter. Figure 2 shows a typical moderate aerosol loading day (e.g AOD < 0.5) over East Asia on December 13$^{th}$ 2018. Figure 2a to 2c are similar to those from Figure 1a to 1c, except this time DB only shows the best quality AOD (QA = 3). The reason that we show only the best quality DB AOD here is that there are sporadic very high AODs scattered in the lower part of the granule, which is clearly contaminated AOD by clouds or other artifacts. Figure 2d and 2f are NDSI value and the brightness temperature of 11 μm, respectively. Both are used in the DT algorithm for masking out snow. Figure 2a shows a thick cloud deck covering the lower part of the granule, aerosol loading is low to moderate over East Asia, which is sufficiently diffuse to allow characterization of the surface cover. There are a couple tiny spots with visual evidence of snow or frozen ponds in eastern China, marked using arrows (yellow or red depending on the background image colors), as well as at the top of the image. From NASA Worldview (https://worldview.earthdata.nasa.gov) we can see that two days before the case study, a snowstorm passed over the North China Plain, leaving snow on the ground that could be seen also on the day before the case study. The sequence of events suggests strongly that the day of the case study would continue to have patches of snow left on the ground, even if the snow patches were not explicitly discernible at MODIS resolution. The ground temperature on this day is above freezing but not too high, thus the snow melting is not very rapid. The nearby AERONET site Xuzhou-CUMT reported AOD at 0.5 μm of 0.3 to 0.6. Figure 2b shows that aerosol loading over this region is around 0.2 to 0.4 over land. Higher AOD, around 0.7, is retrieved over the coastal ocean; however, AOD above the adjacent land is not retrieved. Figure 2c and 2d show that the MODIS DB and MAIAC algorithm retrieved AOD over most of eastern China. However, the two products are not agreeing with each other in data coverage and AOD magnitude over some regions and there is still missing data coverage from both products. Note that the visually identified snow patches are all removed from both products.

We checked cloud and inland water masks, and these filters are not masking the pollution plumes in the DT product in this winter case study. Figure 2e shows the snow mask NDSI, the white colour is when NDSI > 0.2, which is mostly snow overland or water surfaces. The algorithm uses NDSI > 0.01 (purple colour) as the threshold, combined with temperature, to mask out snow. The snow features show in white at center, but the snow edges are a combination of noisy colour pixels from red to blue. Essentially any non-black colour in Fig. 2e will be masked out, given that the corresponding temperature is sufficiently cold. Note the relatively high values of NDSI where the arrows (red or yellow depends on the background) point to a snow feature, as defined by visual inspection in Figure 2a. The problem arises in the large area identified as snow by the mask (within the yellow circle) that is not confirmed from visual inspection in Figure 2a. This large area of misidentified snow differs from the identified snow features in that the NDSI ranges between 0.01 and 0.10, with no spatial connectivity to

very snowy surfaces with NDSI > 0.10. Also, the region in the yellow circle is about 4 degrees warmer than the identified snow features that connect to the major snow fields. The yellow circle temperature is about 277 K, while the identified features are closer to 273 K (Figure 2f). Figure 2 illustrates how the snow mask can falsely mask out moderate to low aerosol loading over winter in China. This is a major reason that DT misses large areas of retrievals over this region during this season.

## 4 Research algorithm for eastern China and a new regional aerosol model

### 4.1 Increasing data coverage by the DT algorithm in China

Based on the case studies, we targeted two major causes of missing aerosol retrievals over winter-time China: inland water mask and snow mask. We then developed a method that can "rescue" the missing retrievals by altering these masks. Previous studies had shown that when NDVI values are between -0.02 to 0.1, the observed scenes can be coastal areas, surface with standing water, arid/desert surfaces, aerosols near cloud edges, and optically thick aerosol plumes (Shi 2018). While we do not want to lose thick aerosol plumes, some of these situations are undesirable for an aerosol retrieval, and we use the NDVI test to mask those scenes. Thus, simply relaxing the NDVI threshold to below 0.1 will likely cause artifacts in retrieved AOD. We need another means to separate desirable from undesirable surface features other than a conservative threshold of NDVI. Reflectance at 2.13 μm is less affected by aerosol and strongly absorbed by water. According to this character, Yang et al., 2020 modify the inland water mask method for the haze conditions by simply adding additional filter $\rho_{2.13}<0.08$ but remaining the NDVI threshold unchanged. The haze aerosol has been successfully retrieved from a MODIS-like sensor MERSI (Medium Resolution Spectral Imager) onboard Chinese Fengyun-3D satellite.

The goal of this paper then is to develop new masking procedures also targeting coastal and semi-arid surfaces, and then relax the NDVI thresholds. We start with reflectance at 2.13 μm and $NDVI_{swir}$ which use the reflectance at TOA at 1.24 and 2.13 μm as shown in Eq. (4).

$$NDVI_{swir} = {\rho_{1.24}\text{-}\rho_{2.13}}/{\rho_{1.24} + \rho_{2.13}} \quad (4).$$

A pixel was determined to be inland water when conditions in Eq. (5) are met:

When $NDVI < -0.02$

or

when $-0.02 < NDVI < 0.1$:

$$\rho_{2.13} < 0.08 \text{ or } \rho_{2.13} > 0.25 \text{ or } NDVI_{swir} < 0.1 \quad 5).$$

where ρ is the reflectance at TOA at 1.24 and 2.13 μm. Empirical investigation determined that $NDVI < -0.02$ is absolute water, when NDVI between -0.02 and 0.1, $\rho_{2.13} < 0.08$ identifies undesirable coastal regions (Yang et al., 2020), and $\rho_{2.13} > 0.25$ or $NDVI_{swir} < 0.1$ identifies semi-arid areas. Pixels within the -0.02 to 0.1 NDVI range, and not caught by these additional filters, are likely due to heavy aerosol loading and should be retained for retrieval.

Snow mask was modified as well to ensure low to moderate aerosol loading can be retrieved. From the case study in Section 3.2 we know that NDSI cannot be modified due to snow edge scenes; however, we can relax the $BT_{11}$ from 285 K to 278 K to exclude false snow detections when NDSI > 0.01. Again, the surprisingly warm temperature threshold was previously chosen to filter out tropical high-altitude snow. To avoid artifacts when temperature is relatively warm and NDSI is very high we exclude pixels with NDSI > 0.2 and $BT_{11}$ < 285 K. We tested this change based on Li et al., (2005) and over global mid to high latitudes and the results suggest that we can apply this new temperature threshold outside of our study region.

### 4.2 Maintaining accuracy for high AOD retrievals

#### 4.2.1 Research algorithm regional aerosol model

These modifications of the inland water and snow masks will increase the data coverage of the DT aerosol retrievals in both thick and thin pollution during winter over East China. There is a potential for a large number of new retrievals, especially at the high AOD end. Because poor assumptions in aerosol optical models are amplified in higher aerosol loading situations, there is a danger that by adding new high AOD retrievals, it will damage the overall accuracy of the product, even if no artifacts are introduced by the change in masking. As we mentioned in Section 2.1, over eastern China, the predetermined regional aerosol models are either moderate absorbing or non-absorbing depending on the region and season (Levy et al., 2007a). These analyses were done more than ten years ago. Now that we may be introducing many additional high AOD retrievals, it is important to revisit the aerosol model choice for our study region (Ichoku et al., 2003), especially since the aerosol environment is undergoing rapid change and there are expanded data sets available to inform the analyses. We develop a local aerosol model by using AERONET (version3, level 2) derived size distribution and complex refractive index from 24 sites, grouped into three clusters and then separated into summer and winter season. See Figure 3.
Figure 4 shows three volume size distributions of 22 particle radii sorted as a function of $AOD_{0.675}$ into bins of 0-0.2, 0.2-0.4, 0.4-0.7, 0.7-1.0, 1.0-1.5, 1.5-2.0, 2.0-3.0, and above 3.0 with the mean of each bin plotted. Figure 4 includes all seasons. Most bins have hundreds to thousands of data points. There is a systematic relationship between particle size distribution and AOD, with fine particle median effective radius ($r_v$) increasing with increasing $AOD_{0.675}$. This relationship appears in all three clusters when ignoring the last AOD bin, which Figure 4b and c only have 15 and 1 data points within these two last bins. The size distribution from Figure 4a and 4b are very similar, especially over the fine mode aerosol regime. The fine mode size distribution of cluster 3 is larger than the other clusters, probably because cluster 3 is warmer and more humid, which leads to larger particle from swelling effects. There are more coarse mode particles in cluster 1 than cluster 2, which could be due to more dust particles in springtime, or coagulation of soot particles in winter from public heating over the northern part of China. To further investigate differences in the aerosol model between winter and summer, we plotted volume size distributions from April to September (summer) and from October to March (winter) over cluster 1 (Figure 5). Figure 5 shows that there is not much difference between the two time periods. The fine mode size distribution is slightly skewed to the right in Figure 5a than that from the Figure 5b, but this is hardly perceptible. As for coarse mode, Figure 5a

has a slightly different shape with a barely noticeable larger amount of coarse mode than Figure 5b, perhaps due to transported dust in spring. Note that although in Fig 5b the size distribution of AOD > 3 (black line) is higher than Fig. 5a over coarse mode region, we refrain from drawing sweeping conclusions about the size distribution of these very heavy aerosol events because the number of data points in this AOD bin are 5 to 10 times smaller than the rest of the AOD bins. Based on the analysis of these plots, we decided to use one averaged size distribution to represent the bimodal pollution aerosol model over northern to eastern China, summer and winter. Note that this fine-dominated pollution model will include the coarse mode, as seen in Figures 4 and 5. During the retrieval, it will be mixed with another bimodal model representing an aerosol dominated by dust (Levy et al. 2007a, b).

Figure 6 shows the spectral dependence of the real and imaginary parts of the refractive index for all inversions and sorted as a function of $AOD_{0.675}$. Because the AERONET inversion does not report a complex refractive for each inverted size distribution, to increase the sample size in each AOD bin, we only use 3 AOD bins (0.4-1.0, 1.0-2.0, 2.0-4.0). Based on the AERONET teams' recommendation, only AERONET refractive index values, with corresponding $AOD_{0.44}$ larger than 0.4 are used to generate the model (Holben et al., 2006). Figure 6 shows, there is no systematic relationship between the real part of the refractive index and AOD in this data set. The variability in each AOD bin exceeds the differences between the bins. There is a slight separation between low AOD bin vs. high AOD bin in the imaginary part of the refractive index. Thus, we use a single mean value for the real part of the refractive index and a parametric equation based on AOD for the imaginary part of the refractive index in our regional aerosol model. The real part of refractive index is interpolated to 0.55 µm linearly, while the imaginary part of the refractive index is interpolated using logarithms from 0.44 µm and 0.675 µm (Lee et al., 2017).

Table 1 shows the comparison between the fine modes of the operational models that are used over the China region and the newly generated aerosol model. The coarse modes remain the same as the operational models (Levy et al., 2007b). The calculated natural logarithm of the standard deviation of the radius (σ) and the volume of particles per cross section of the atmospheric column ($V_0$) don't change much from the operational non-absorbing model. And our sensitivity studies show that changes in these two parameters are not the major factors in changing the output AOD. Thus, these two parameters remain the same. The new modal radius is very similar to what has been used operationally over part of coastal China during Fall and Spring seasons, namely the "non-absorbing model" in Table 1. However, we are extending the same size distribution to a larger area of China over winter. The differences in the imaginary part of the refractive index show that when compared with the non-absorbing model, the new model is more absorbing in the low AOD range but less absorbing when AOD > ~2. When compared with the operational moderate absorbing model, the regional model is more absorbing when AOD < 0.5 but are less absorbing when AOD is greater than this value. Overall, the new aerosol model is in between the operational non-absorbing aerosol model and moderate absorbing model when aerosol loading is moderate. Also notice that the moderate absorbing aerosol model shows increased absorption with increasing AOD, which is opposite to the non-absorbing model as well as to the regional model. This indicates that with increase of aerosol loading, the absorption

decreases in this region. These differences, especially due to the differences in absorption can introduce a retrieval bias in AOD on the order of one.

**Table 1: Optical properties of the aerosol model used by the operational DT algorithm over China and the regional model generated in this study using AERONET inversion products. Real and imaginary refractive index is a spectrally dependent quantity. Values in this table are for 0.55 µm.**

| Model | $r_v$, µm | σ | $V_0$, µm³/µm² | Real part of Refractive Index | Imaginary part of Refractive Index |
|---|---|---|---|---|---|
| Non-absorbing | 0.043τ + 0.160 | 0.1529τ + 0.364 | $0.1718\tau^{0.821}$ | 1.42 | 0.0015τ-0.007 |
| Moderate absorbing | 0.020τ + 0.145 | 0.1365τ + 0.374 | $0.1642\tau^{0.775}$ | 1.43 | -0.002τ-0.008 |
| Regional | 0.046τ + 0.11 | 0.1529τ + 0.364 | $0.1718\tau^{0.821}$ | 1.49 | 0.0033τ-0.011 |

### 4.2.2 Aerosol layer scale height

Other than creating a new model, we also adjusted the aerosol layer height assumption when generating the LUT. In the operational algorithm, the LUT is calculated using an aerosol layer scale height of 2.0 km. Over China, when high loading of pollution accumulates there is usually a high-pressure synoptic system, which suppress the aerosol layer vertical height (Zhao et al., 2013). Many field measurements reported the planetary boundary layer height during pollution episodes near the Beijing area to be only 800-1000 meters (Tang et al., 2015, Li et al., 2015, Luan et al., 2018). A 4-year climatology of CALIOP aerosol layer height over wintertime China is also around 1.0 -1.3 km over northeastern China (Liu et al., 2015). Thus, the scale height of the pollution layer in China is set to be around 0.5 km, which means that 80% of the aerosols are within 1 km of the surface. This new assumed height is much lower than the operational value of 2 km, but we know that heavily polluted conditions increase the atmosphere stability and reduce the boundary layer height (Petäjä et al., 2016, Miao et al., 2017).

The DT algorithm has never changed aerosol scale height in the two decades of its operational history. The value is hard-wired into the LUT calculation, as are other assumptions such as particle size distribution, refractive indices and shape. Unlike aerosol retrieval algorithms that make use of measurements in the UV part of the spectrum (Torres et al., 2012), the DT algorithm relies only on visible and SWIR wavelengths, which are less sensitive to variations in scale height than the UV-dependent algorithms. Besides, aerosol layer height is variable on short temporal and spatial scales, making adjustments to the global constant value difficult to implement operationally. However, differences in aerosol layer height can impact retrieved AOD especially for more absorbing aerosols. Figure 7 illustrates of how much change to expect in AOD if aerosol scale height changes from 2 km to 0.5 km using the moderate absorbing model. The blue and red curves denote the calculated reflectance at TOA for a range of AODs, with blue representing aerosol at 0.5 km scale height and red for 2.0 km scale height. For a measured reflectance of 0.3, the AOD for the 0.5 km scale height would be 2.5, while for the 2.0 km scale

height it would be 3.0. For a measured reflectance of 0.27, the AODs would be 1.8 and 2.0, respectively. Figure 7 shows for an aerosol model whose single scattering albedo is 0.92, when AOD is around 2 to 3, the percentage differences in TOA reflectance between the two scale heights is ~ 3-5%, which leads to AOD changes of ~8-15%. Similarly, for the non-absorbing model (SSA = 0.95), the changes in AOD is around 4-7% when AOD is 2 to 3. Those are significant changes that require consideration and might be addressed for our specific region of interest as we develop a regional aerosol model for this research algorithm. We note that even adjusting the aerosol scale height for our specific region in certain conditions may improve retrievals in those conditions but make things worse at other times, as aerosol layer height varies temporally. We choose to optimize for high AOD conditions, accepting the possibility that biases may be introduced when AOD is low.

Using the regional pollution model with reduced aerosol scale height of 0.5 km and the algorithm with modified masking, we re-produce cases 3.1 and 3.2, as shown in Figure 8. Figure 8 a-c are for case study 3.1 (9 October 2013) where Figure 8a shows the MODIS operational DT AOD with applied quality assurance (QA) equal to the highest value (=3), Figure 8b shows the AOD produced by the regional research algorithm with QA=3, and Figure 8c gives the differences in AOD (Figure 8b minus 8a). The red-blue color scale is the AOD differences, with the extra data coverage in Figure 8b highlighted in green. Comparing Figure 8b with Figure 8a, AOD are retrieved now at the center of the plume, where the operational DT has failed to retrieve. The AOD values change from 1.0 near the edge of the plume to ~5 at the center. The pattern of the plume fits what we see from the RGB image (Figure 1a). The difference plot shows that with the aerosol model and scale height change, the change in AOD is mostly less than 0.1, most of which is increasing AOD. The increase in AOD is mostly due to the aerosol model change while the decreasing AOD is probably due to the aerosol scale height change. Figure 8 d-f are similar to Figure 8 a-c but for case 3.2 (13 December 2018). Figure 8e has much increased data coverage over eastern China with AOD values less than 1.0. Figure 2 shows these areas have no cloud or snow cover, and aerosol loading is generally less than 1.0, which fits what the new algorithm has retrieved. The difference plot mostly shows larger amounts of new pixels which are not retrieved in the operational algorithm. The change in AOD is mostly positive, and less than 0.06. The reason for the large area of no data over the northern part of China is due to the surface being bright in the 2.1$\mu$m reflectance, which causes the retrieved AOD to be assigned to a lower QA value, and thus was not shown in Figure 8d and 8e where QA is required to be equal to 3.

## 5 Validation of the research AOD for January to March 2013

The research algorithm is applied to MODIS radiances in a region bounded by 100° E to 130° E and 20° to 42° N, during January to March 2013. The resulting AOD are evaluated against AERONET AODs and inter-compared with AODs from the operational DT product. Spatiotemporal collocations of MODIS retrievals within 0.3° Lat/Lon of the AERONET site location and AERONET observations within 30 minutes of the satellite overpass times are used to collocate the two data sets. Figure 9 shows the scatter plot of MODIS versus AERONET AODs for (1) the operational DT product, (2) AOD retrieved using modified masks with the operational LUT, (3) the research version of the MODIS AOD using the new LUT

and the modified masks (referred to as the research algorithm hereafter), and (4) the new points that appear in (3) but were not there in (1), along with the error statistics and error envelopes (±0.05±15%AOD) determined from the global operational DT product (Levy et al., 2013). Figure 9a shows that the MODIS DT product correlates well with AERONET data with 60.5% of collocations falling within the expected error (EE). However, we can see there is almost no data greater than 2.0 that are reported when collocated with AERONET. Comparing the effect of the new masking (Figure 9b) with the operational data, the biggest change is that the total number of AOD collocations increased 50% and the number of AOD > 1 almost doubled, although these increases in high AOD also increases the root mean square error (RMSE) and reduces the percentage of data within the EE. This jump in RMSE is halved and the high bias is reduced in Figure 9c, which uses the new LUT. However, the model change also leads to additional low bias at low AOD when compared with the operational product, which is linked to optimizing the scale height change for heavy aerosol loading. The overall bias is very small for the research algorithm, partially because there are both high biases and low biases within the newly added high AOD, which leads to a mean bias of almost zero. But nevertheless, the data coverage has increased, especially over the high AOD regime. Figure 9c shows that there are almost 300 new pairs of MODIS-AERONET collocations found in the research product. The newly added data show similar validation statistics as the rest of the data (Figure 9d), which indicates there is minimum snow contamination or other artifacts impacting these data.

We analyze the satellite-AERONET bias of the DT and research AOD as a function of AERONET AOD and show the results in Figure 10. In Figure 10, AOD is binned every 47 pixels. When AERONET AOD is less than 1.0, there is small bias between all MODIS products and the AERONET AOD. When AERONET AOD is greater than 1.0, the negative bias in the DT AOD grows to around -0.1 and then to -0.3 when AERONET is around 2.0. The mean negative bias in the C6 AOD at AERONET $AOD_{0.55}$ > 1.0 is partially due to the generic aerosol model used in the operational algorithm that is more absorbing than the heavy pollution generated in wintertime over eastern China. The AOD retrieved with the operational LUT but modified filters include more collocations at high AOD, which leads to larger positive bias. The research product maintains an absolute mean bias against AERONET of 0.01 or less across the entire range of AERONET AODs and shows very good agreement at the very highest AODs ($AOD_{0.55}$ > 2.3). The very small mean bias is partially due to cancellation effects of overestimation and underestimation of research AODs as shown in Figure 9c. The standard deviation of the bias can be large even when the mean bias is low. The regional aerosol model we use represents the fine mode aerosol over the majority of China except the west and center-northern part of China, where other aerosol types can occur.

The new research algorithm increases data coverage temporally and spatially. A daily averaged AOD time series at the Xianghe AERONET site and the MODIS operational and research aerosol product is shown in Figure 11. The Xianghe site is located near Beijing, where many heavy pollution episodes occur. Thus, the AOD time series over this site show the most significant differences in data coverage between the DT AOD and the research AOD. The time series covers from January to March 2013. When AERONET observes AOD < 0.5, both the operational DT and research products capture the AOD equally well. However, when AERONET observes AOD > 1., the operational product fails to retrieve the AOD while the research product obtains a lot more AOD values over these days. Part of the discrepancies between the ground based and

satellite measurements are due to sampling differences as well as ground conditions such as snow cover or melting snow. For example, AERONET AOD around 2.7 at Julian day of 15 is reduced to around 1.0 if we restrain the AERONET observation time to 30 mins before or after the MODIS passing time. Similarly, the overestimation of research AOD above 3 between Julian day of 75 and 80 is much smaller if we use the collocated data set instead of the daily average.

To further investigate the ability of using the DT research product to identify the pollution events, we calculated the AERONET-identified pollution day using three AERONET sites, Beijing, Beijing-CAMS, and XiangHe. As long as there are two observed AERONET AOD > 1.0 within one day, that day is considered a polluted day. The number of polluted days identified at these three sites between January and March 2013 are: 19, 16, and 23, for Beijing, Beijing-CAMS, and XiangHe, respectively. There are also sampling differences between the three AERONET sites even though they are within 1° latitude and longitude of each other. Between Beijing and Beijing_CAM, there are 10 identified days in common. Between Beijing (Beijing_CAM) and XiangHe, there are 14 (12) identified days in common. Among all three sites, there are only 7 days that are commonly considered polluted day. Identified pollution event days are listed in Table 2.

The research product identified a total of 39 polluted days, of which 22 days were also identified as polluted by at least one of the three AERONET sites. There were 17 days when the research product identified a polluted day but AERONET did not, and 7 days when AERONET observed AOD > 1.0 but the research algorithm did not capture the event. It is easy to understand when AERONET identified a polluted day but the research retrieval did not, because the AERONET observation time can be different from MODIS overpass time. The polluted scene can be cloud covered at over pass, but be captured by AERONET before or after, or the scene can significantly change between two observing times. It is more difficult to understand how the research algorithm could identify a pollution event on 17 days that all three AERONET stations missed. To confirm polluted days that the satellite identified but AERONET did not, we visually compared each day using RGB images and MODIS DB and MAIAC AOD retrievals. Among these 17 days, 14 days have dense pollution present visually (with retrieval over cloud free/snow free land or ocean). The spatial coverage offered by the satellite was able to pick up pollution events missed by ground stations, even when the ground stations were densely sited according to global network standards. The 3 days identified by the satellite as pollution events but could not be confirmed by visual inspection were cloudy. In these three cases we expect cloud effects in the MODIS data that do not appear in the AERONET data caused the AOD to exceed the AOD = 1.0 threshold. We note that none of the three days in question have AOD over visually identified snow patches. Overall, we are happy with the ability of using the DT research product to identify pollution events, which can be more efficient than using sparse ground observations.

**Table 2 AERONET and DT research product identified days with pollution events from January to March 2013 over the Beijing area (37-40 N and 115-118 N).**

| Months | Beijing | Beijing_CAM | XiangHe | Research |
|---|---|---|---|---|

| | | | | |
|---|---|---|---|---|
| Day of Year | 11,14,19,27,28,42,44,47,52,65,66,67,68,70,73,74,75,76,80 | 11,14,15,19,22,27,28,73,74,75,76,80,84,85,88,89 | 7,10,15,18,19,21,27,42,44,47,66,67,68,70,73,74,75,76,80,84,85,88,89 | 7,10,11,12,13,15,19,22,36,40,41,44,45,47,49,51,52,54,55,57,58,61,64,65,66,67,68,70,73,74,75,76,78,80,82,85,86,88,89 |

**6 Characterization of the 2013 winter China pollution situation**

With the research algorithm able to make many more additional retrievals and produce a better representation of the aerosol during winter, we examine the aerosol situation over China from January to March 2013 and investigate how the new results differ from the operational in characterizing this situation. Figure 12 shows the AOD distribution change from operational DT AOD to research AOD in log-scale. The red is the research AOD and the blue is the operational AOD. The histogram shows that when AOD is less than 1.0, the number of AOD retrievals increase about 5%-6%. When AOD is greater than 1.0, and especially greater than 2.0, the increase in number of retrievals is much larger. The number between AOD 1.0 to 2.0 increased 47% while after that the number of data points doubled or tripled. In the negative AOD bin the number of data points is the same between the data products, thus, no red bar can be seen.

Monthly mean domain-averaged AOD statistics are shown in Table 2 for both MODIS aerosol retrievals over land. The operational DT product shows similar averaged AOD values around 0.58 and number of retrieved pixels around 18K in January and February. Both the DT AOD value and number of retrieved pixels increased in March. Compared to the DT AOD, the research AOD is higher and the difference is largest in January (~0.17 in AOD) and smallest in March (~0.11). The same patterns can be found in the number of pixels, the increment is about 40K in January and 12K in March. This means that many more heavy pollution episodes occur, and possibly were missed by the operational DT algorithm, in January than in the other two months. There is a reduction in research AOD in February, which could be caused by increasing the number of retrieved AOD smaller than 0.5, or by lowering the aerosol scale height without adding new retrievals. February is also associated with a decrease in the number of retrievals from the numbers seen in January, which could be caused by increased snow or cloud cover in February 2013.

**Table 3 Domain-averaged (25° to 40° N and 105° to 120° E) monthly mean MODIS-derived AOD with QA=3 at 0.55 $\mu$m over land for the operational (DT) and research (Res) algorithms and the number of valid retrievals in 2013.**

| Months | DT Land AOD | Res Land AOD | DT Land # pixels | Res Land # pixels |
|---|---|---|---|---|
| January | 0.561 | 0.732 | 18541 | 64070 |
| February | 0.584 | 0.704 | 18589 | 36610 |
| March | 0.662 | 0.778 | 80064 | 92674 |

Figure 13 shows the spatial distribution of averaged AOD from the research product and the operational product at 0.5° resolution over the study domain from January to March 2013 with at least 3 data points per month per grid. The upper row is the operational DT AOD and the lower row is the research AOD with three columns representing January to March. The research product shows much more intense aerosol loading over eastern China with more data coverage north of 35° N, than does the operational DT algorithm. The differences between the monthly DT and research AOD distribution are shown in Figure 14 (upper panels) along with the number of pixel differences (lower panels). Grid boxes with AOD differences greater than 1.0 are found closer to Beijing and its surrounding area. Such large differences are found in 10%, 17%, and 5% of the total land grid boxes in January, February, and March correspondingly within the domain. Near the Beijing area (~40º N, ~116.5º E), differences in gridded AOD can be above 3.0 in March. There is a large number of additional retrievals in the area south of Beijing, bounded by 30° to 35° N and 110° to 120° E, where for each grid the research algorithm produces more than 100 new data points in January and around 40 to 60 in February and March. Although not shown in this paper, we also see increments in the number of data points over northern India and the islands of Japan in January and February. So even though the research algorithm was developed and tested only for China, from the magnitude of increased AOD in these places, there is indication that heavily polluted cases also may be missed over these regions. Non-Chinese locations will require separate validation analysis and would likely benefit from an evaluation and adjustment to the aerosol model used in the LUT. For example, the strong decrease in AOD over southern China seen in Fig. 14 has not been validated and may indicate the local nature of the aerosol model or assumption of aerosol scale height in the research algorithm developed for the Beijing area. Another promising development seen in in Figures 13 and 14 is that there is no increase in the number of data points or significantly increased AOD over coastal regions or over arid and semi-arid area in northern and western China (part of these area is shown here). This indicates that the changes we made to the masking algorithms have not allowed improper surface types to be retrieved.

## 7 Summary and Conclusions

The MODIS DT algorithm misses many retrievals over eastern China during the wintertime when compared with ground-based measurements. Two conditions can lead to missing retrievals, one is during heavy pollution events, and another is in low to moderate aerosol loading when the snow mask is mistakenly invoked. Other than missing retrievals, there is also improvement that can be made to more accurately represent aerosols over this region. Other satellite aerosol products also have trouble representing the scale of the aerosol loading over the study period and domain.

To improve the data coverage without damaging the retrieval accuracy, we adjust the pixel selection routines, specifically the inland water mask and the snow mask. Then we use AERONET version 3 inversion products to first evaluate the aerosol models used operationally in China and then develop an aerosol model specifically for this region. The inland water mask is relaxed to allow for very high AOD, but then used in combination with the reflectance at 2.13 µm to eliminate artifacts from

coastal and brighter surfaces. The snow mask is also modified to include scenes currently misclassified as snow by lowering the threshold of surface temperature during snow cover and snow melting conditions. These measures increase the number of retrievals in our domain by 50% and double the number of retrievals with AOD greater than 1.0. The higher the AOD, the more sensitive the retrieval will be to the aerosol model. After adding so many new high AOD retrievals, we find that a new aerosol model is needed, which we develop from local AERONET inversion products. The new aerosol model has absorption in between the non-absorbing and moderate absorbing models that were used in the operational model. The assumed aerosol layer height was also lowered in the new LUT to match the aerosol vertical distribution over the study region. The combination of new aerosol model and lowered scale heigh reduces high bias for retrievals at high AOD but also introduces low bias at low AOD. That low bias may be accentuated as the algorithm is applied beyond the local Beijing area, or when aerosol conditions change temporally in the local Beijing area. This is the first time that aerosol layer scale height has been adjusted in the DT retrieval, since the at-launch algorithm 20 years ago, and suggests there could be sensitivity to aerosol layer scale height in other regions with heavy aerosol loading.

We validated the research product from January to March 2013 using AERONET version 3 level 2 AOD. With the large number of new AOD retrievals, particularly new high AODs, the RMSE increased and the percentage within the expected error (EE) was reduced by 4%; however, the overall bias was reduced to -0.009. On average, 56% of the collocated research AOD are within the error bounds determined from global analysis of DT retrievals, which is less than optimal. If we relax the error envelopes from 0.05+15%AOD to 0.08+17%AOD then 70% of the research AOD fall within these bounds. These relaxed error bounds represent our best estimate of the uncertainty in the research product for this area during winter.

The ability to now retrieve these optically thick pollution events alters our understanding of the aerosol system in this region. Statistical analyses illustrate the increase of the regional aerosol distribution during wintertime over eastern China, including a very large increment in AOD over Beijing. Using the new algorithm, the monthly-regionally averaged over-land $AOD_{0.55}$ over the domain increases by 0.11 to 0.18 over values calculated from the operational DT products during January to March of 2013, with the largest increment happening in January. But near Beijing where the severe pollution occurs, the new algorithm increases $AOD_{0.55}$ by as much as 3.0 for each 0.5° grid box, over the previous operational algorithm values.

The large area of missing data and the magnitude of missing AOD will heavily alter our understanding of the severity of these pollution events, influence regional radiative balance, and impact the air quality community. Being able to bring back these missing data especially in a near real time manner can significantly influence the aerosol and air quality modeling and forecasting studies as well as any decision making that rely on instantaneous satellite aerosol data. Being able to include these rare but important heavy aerosol events in the DT products is critical to preparing the DT product to be more suitable for a wider range of applications. There is also potential to apply the research algorithm globally especially over regions that high aerosol loading events (e.g. large-scale wildfires or severe air pollutions) frequently occur, such as western U.S and Indian. The modification of inland water mask and snow mask have been lightly tested globally. Results show that inland water mask change introduced differences in retrieved AOD over coastal and arid/semiarid region and snow mask works

well on the selected scenes globally including tropical high mountain regions. However, tests with longer time span are needed before we can commit these changes globally.

## 8 Data Availability

The MODIS level1B reflectance (DOI: 10.5067/MODIS/MYD021KM.061) () and Dark Target level2 aerosol data (DOI: 10.5067/MODIS/MOD04_L2.061) can be accessed via LAADS DAAC (https://ladsweb.modaps.eosdis.nasa.gov/). The AERONET direct sun measurements data used in this study is available via AERONET website (https://aeronet.gsfc.nasa.gov/) (Giles et al., 2019). The DOI of the data is https://doi.org/10.5194/amt-12-169-2019.

## 9 Author Contribution

Yingxi Shi, Robert Levy, and Leiku Yang initiated the research. Yingxi Shi leaded this research and performed the research algorithm development and results analyses. Robert Levy and Lorraine Remer provided guidance throughout this research, especially on algorithm development and aerosol layer height. Leiku Yang created the new inland water mask and helped on case study. Shana Mattoo helped algorithm modifcation implementation. Oleg Dubovik provided knowledge on creating new aerosol model. All authors revised manuscript.

## 11 Competing Interests

The author claims there is no competing interests regarding this paper.

## 12 Acknowledgement

We thank Terra/Aqua senior review and MeASURes (NNH17ZDA001N-MEASURES) provided by National Aeronautics and Space Administration (NASA). We also acknowledge NASA Cooperative Agreement 80NSSC20M0209 provided by the PACE Science and Applications Team under the direction of Laura Lorenzoni and Paula Bontempi. The part of Dr. Yang's work is supported by the National Natural Science Foundation of China (NSFC, 41975036). The authors thank the AERONET team for establishing and maintaining the AERONET sites and the data used in this investigation.

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

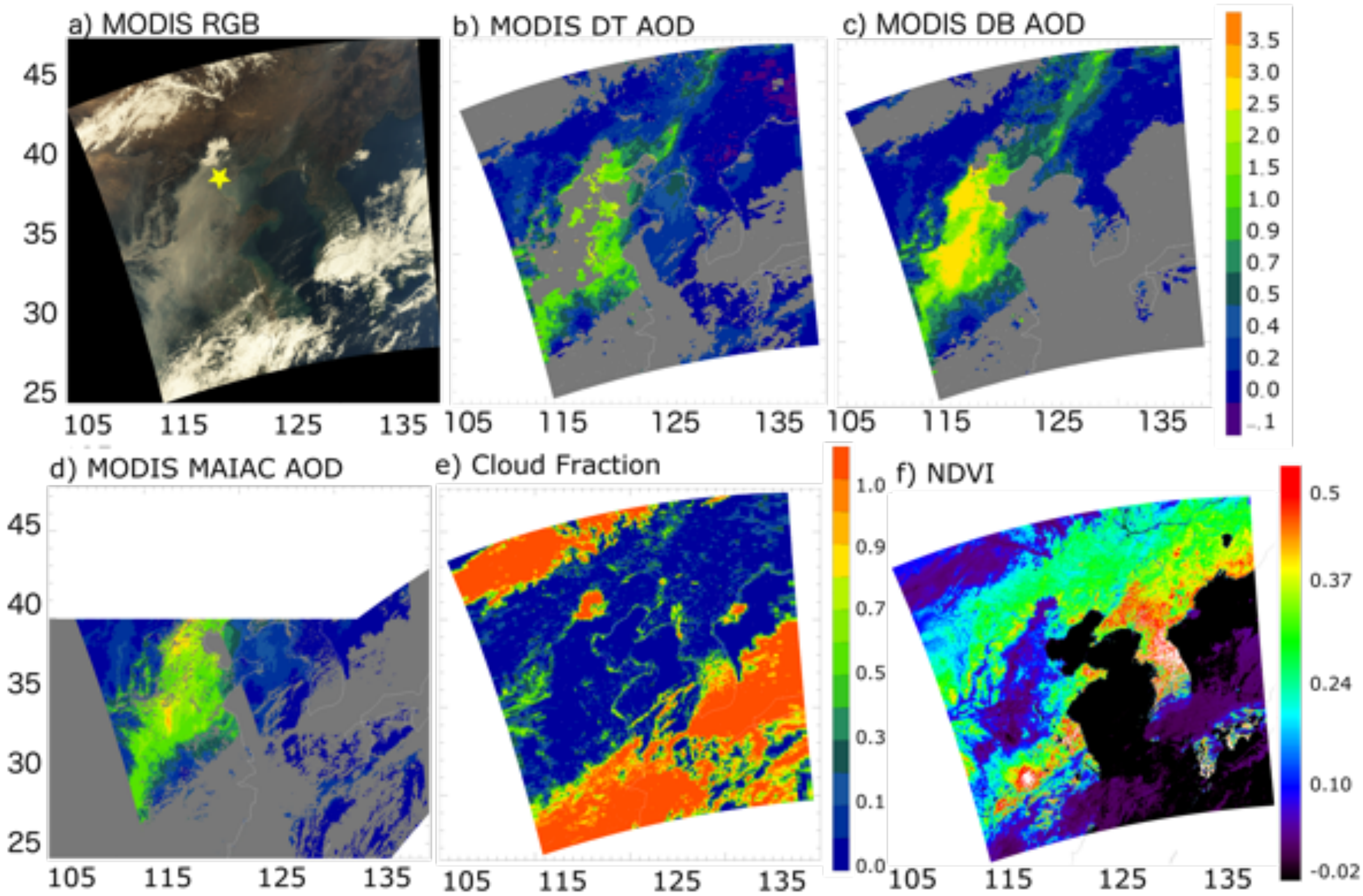

**Figure 1 A case study of a pollution event over eastern China on 9 October, 2013. a) MODIS Aqua RGB image, b) MODIS Dark Target (DT) AOD at .0.55 µm for all available retrievals (QA = 0 to 3), c) MODIS Deep Blue (DB) AOD at 0.55 µm, for all available DB retrievals (QA = 0 to 3) d) MODIS MAIAC AOD e) MODIS DT Cloud Fraction , a diagnostic of the MODIS aerosol product, f) Normalized Difference vegetation index, used as the inland water mask by the DT algorithm . The yellow star represents the AERONET site Xianghe.**

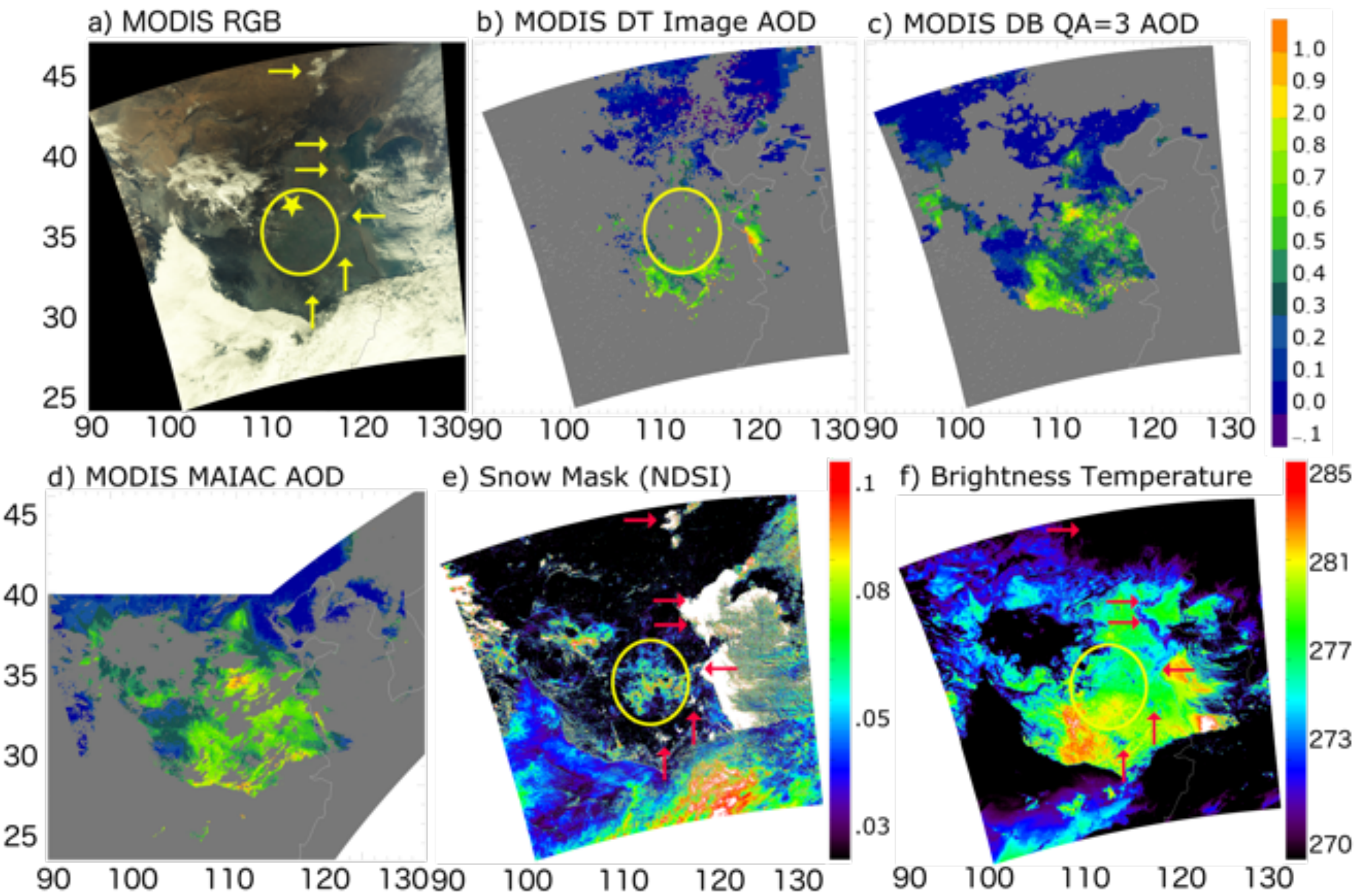

**Figure 2 A case study of a pollution over Eastern China on 13 December 2018. a) RGB image, b) MODIS DT all available AOD (QA = 0 to 3), c) MODIS DB AOD with QA=3, d) MODIS MAIAC all available AOD, e) MODIS DT NDSI used for snow masking, f) 11 µm brightness temperature. The yellow star represents the AERONET site XuZhou-CUMT. Arrows point at the snow patch locations in RGB, NDSI, and BT images, and yellow circle encompasses the problem region.**
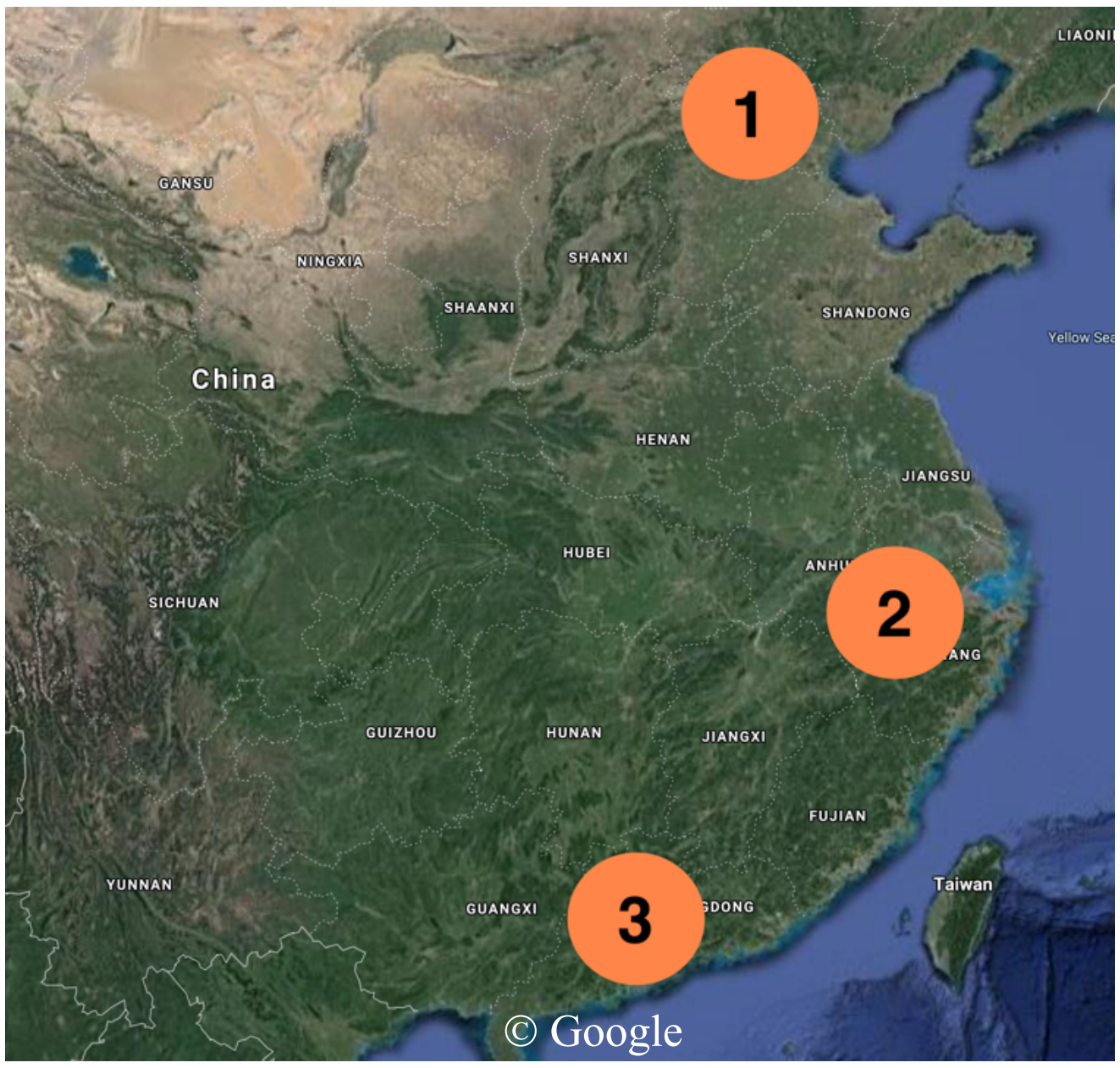

785 **Figure 3 Clusters of AERONET sites that are used in this study to generate the regional aerosol model. Cluster 1 includes: Beijing, Beijing-CAMS, Beijing-RADI, Beijing_PKU, Lingshan_Mountain, Liangning, PKU_PEK, XiangHe, Xinglong, Yufa_PEK, and Yanqihu. Cluster 2 includes: Hefei, NUIST, Shouxian, Hangzhou-ZFU, Qiandaohu, Hangzhou_City, Taihu. Cluster 3 includes: Hong_Kong_Hok_Tsui, Hong_Kong_PolyU, Hong_Kong_Sheung, Kaiping, Zhongshan, Zhongshan_Univ. The background map is from © Google map.**

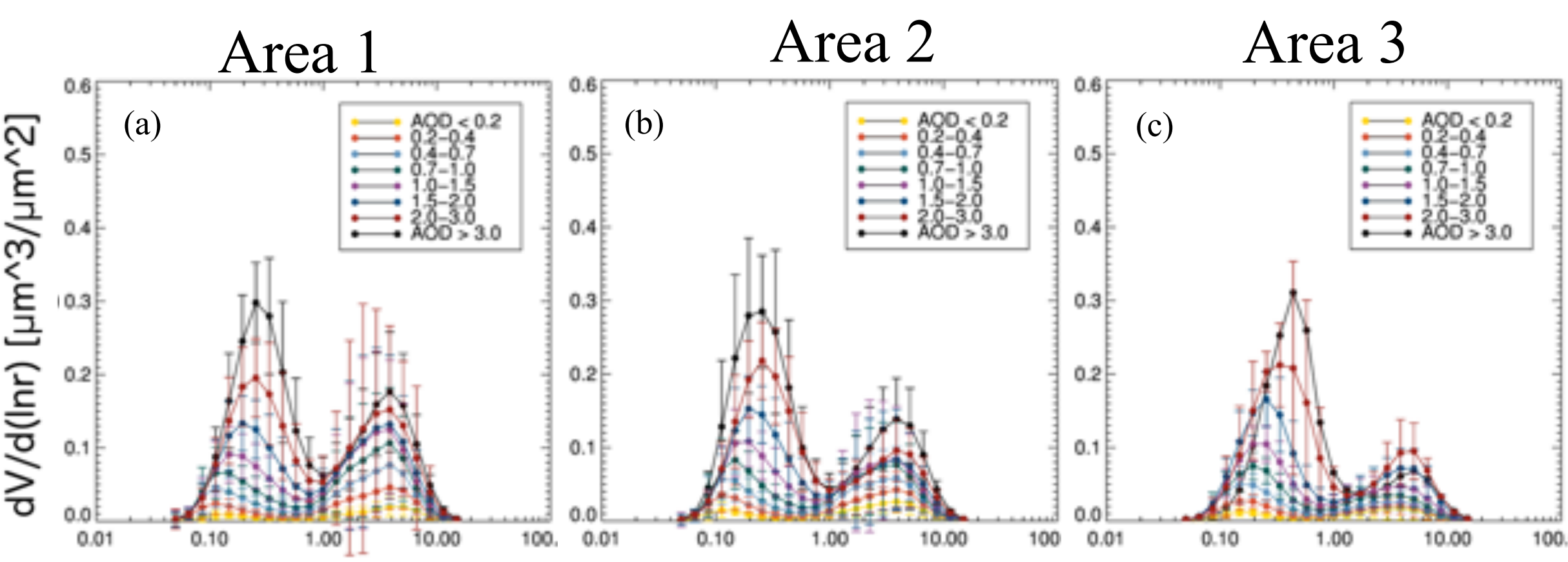

790 **Figure 4: Size distribution as a function of AERONET AOD at 0.675 µm, generated from the AERONET inversion products at three clusters illustrated in Figure 3 using all available data records. Other than the last AOD bin, which is AOD > 3, the number of retrievals within each AOD bin are between hundreds to thousands. There are 263, 15, and 1 data points in the highest AOD bin for cluster 1, 2, and 3, respectively. The error bars represent the standard deviation within each size bin**

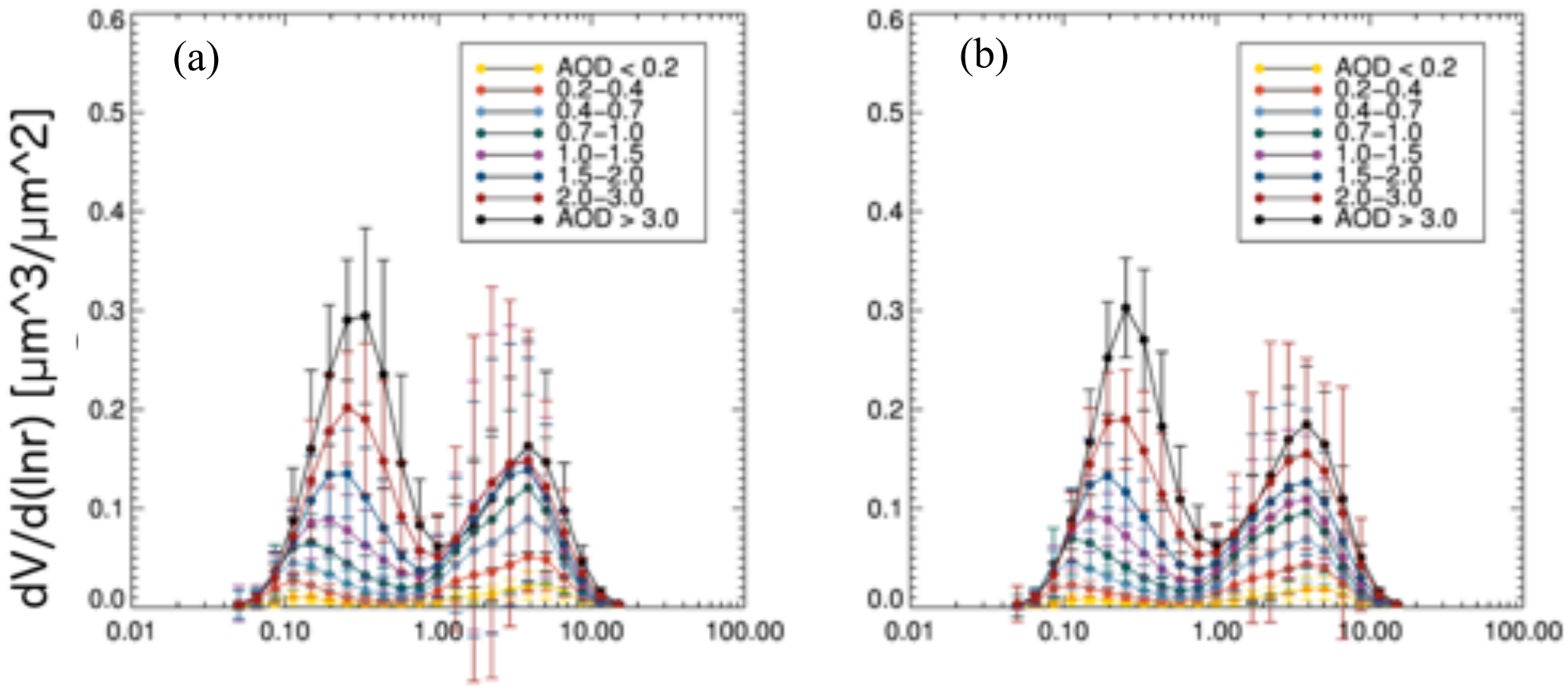

**Figure 5: Size distribution as a function of AERONET AOD at 0.675 µm, generated from the AERONET inversion products at cluster 1 illustrated in Figure 3 using all available data records, a) is from April to September, b) is from October to March. The error bars represent the standard deviation within each size bin.**

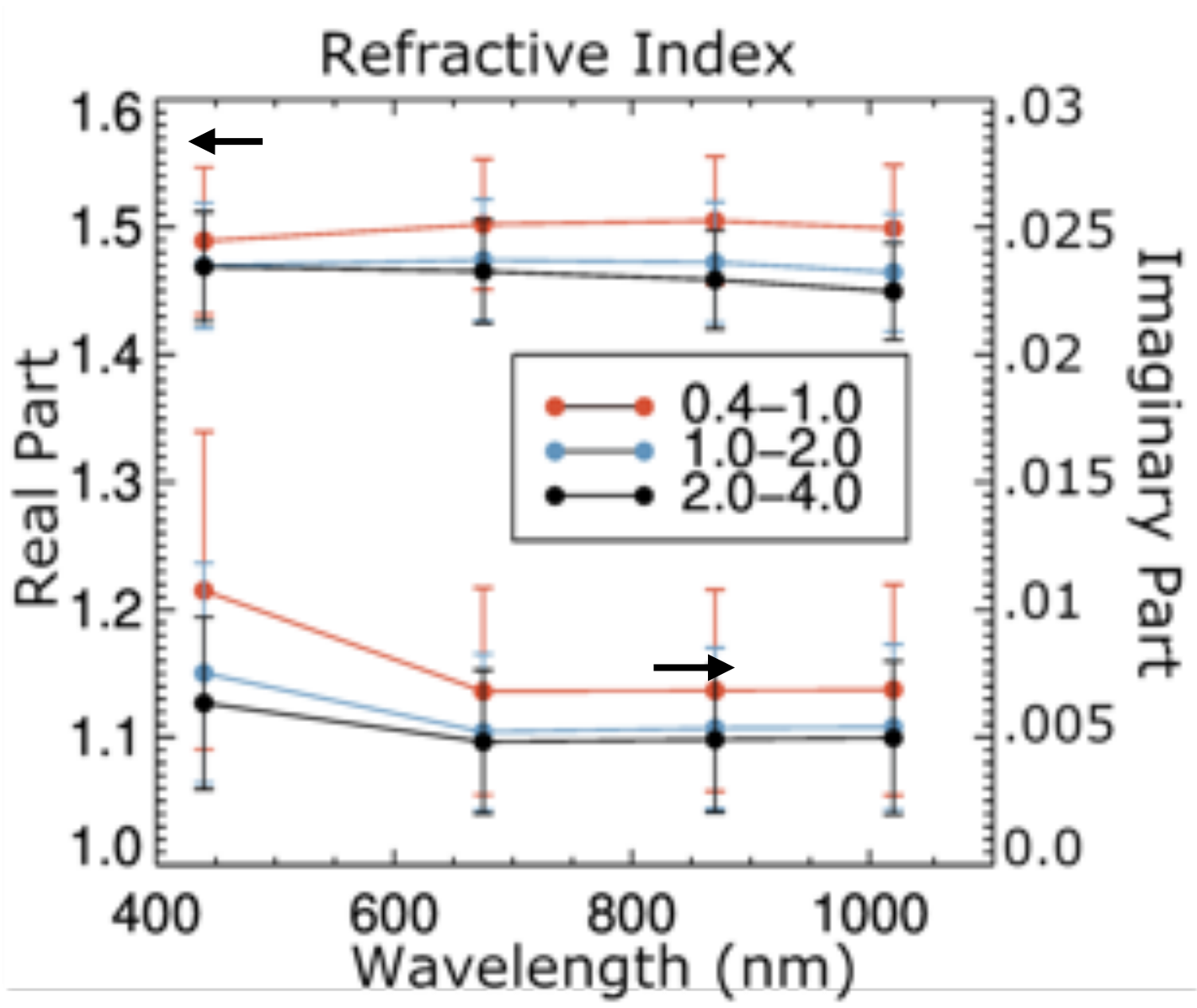

**Figure 6 The real (top set of curves and left axis) and imaginary (lower set of curves and right axis) parts of refractive index as a function of AOD at 0.675 µm calculated from the AERONET inversion product at cluster 1. The error bars represent the standard deviation within each wavelength.**

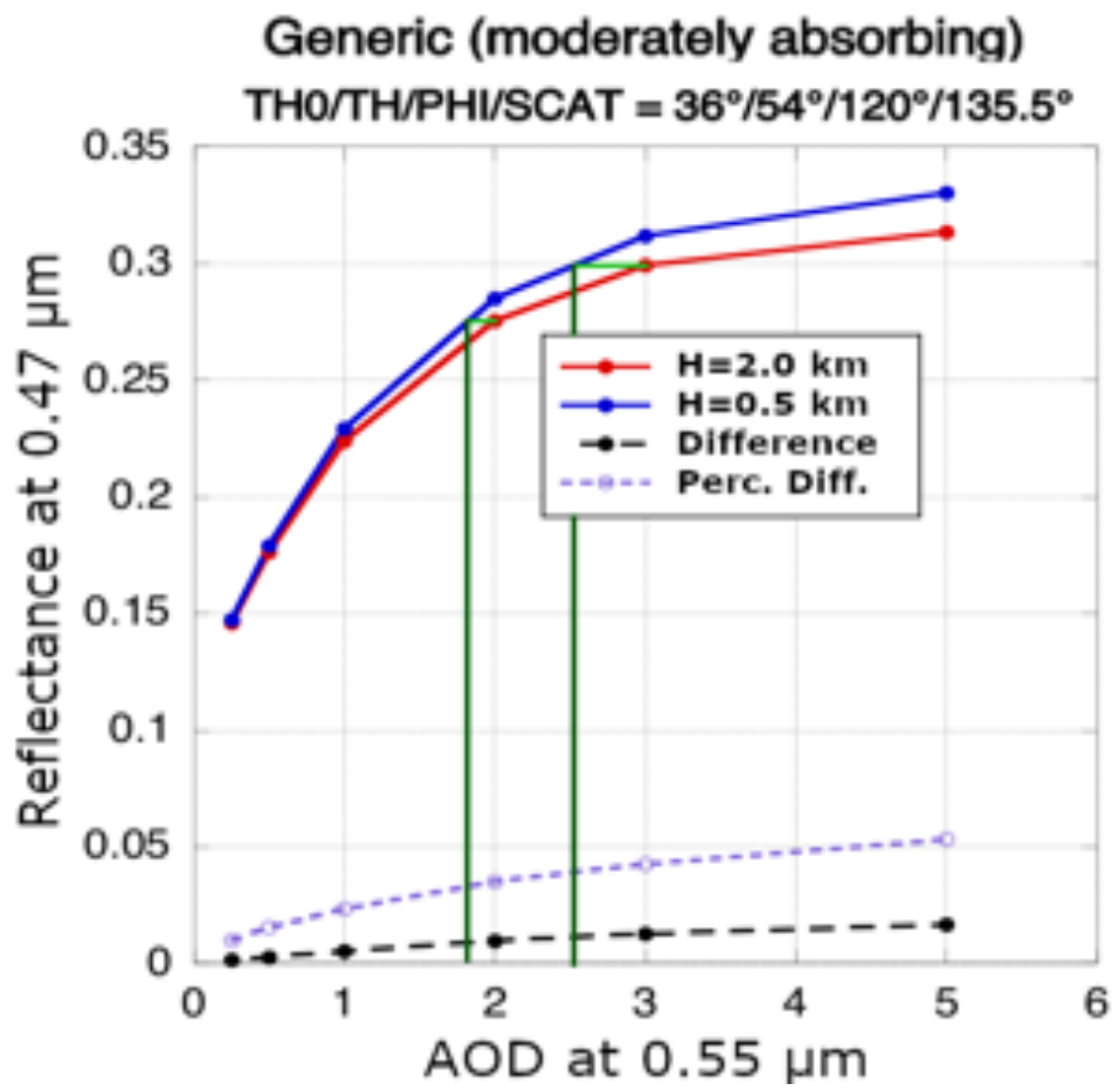

**Figure 7 Top-of-atmosphere reflectance at 0.47 µm corresponding to AOD at 0.55 µm using the moderate absorbing model for the specific geometry of solar zenith angle = 36°, view zenith angle = 54°, relative azimuth angle = 120° and scattering angle = 135.5°. The red and blue lines are the calculated TOA reflectance at aerosol scale heights of 2.0 and 0.5 km, respectively. The black and purple dashed lines are the differences between the red and blue line and the percentage differences. The light green horizontal line segments indicate TOA reflectance when AOD are 2 and 3 using scale height of 2.0 km. The corresponding dark green vertical lines are corresponding AOD using scale height of 0.5 km.**

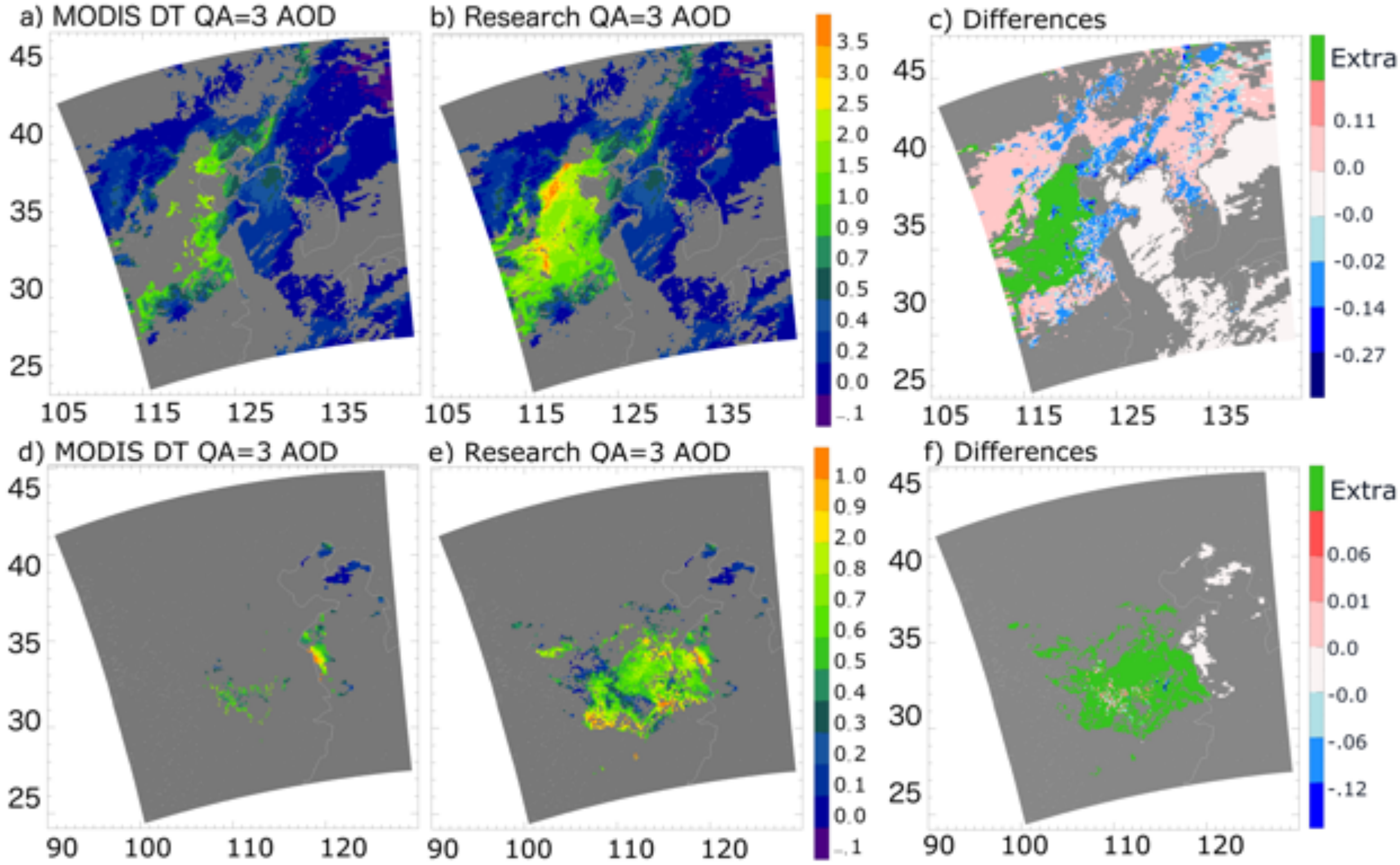

**Figure 8 a) to c) AOD images for the case study over Eastern China on 9 October, 2013: a) operational DT AOD at 0.55 µm with QA = 3, b) Research AOD at 0.55 µm with QA = 3 using altered thresholds on the NDVI test, snow test, and a new regional aerosol model with new aerosol scale height. c) The differences between the research AOD (panel b) and the DT AOD (panel a). d) to f) are for a case study of moderate pollution over Eastern China on 13 December 2018. The increased research AOD data coverage is shown in green.**

Atmospheric Measurement Techniques
EGU
Discussions

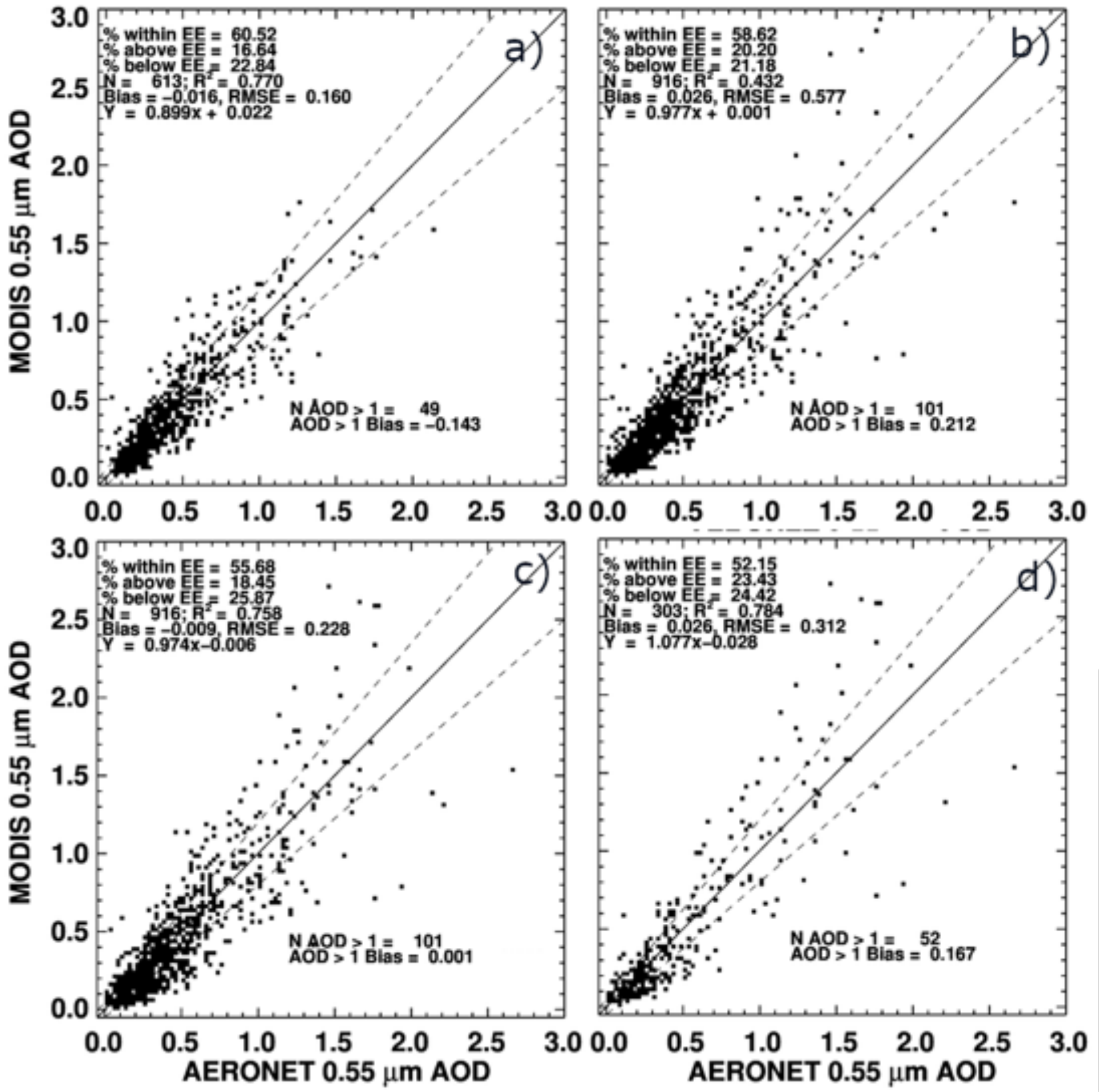

**Figure 9 Comparisons of the MODIS DT AOD at 0.55 µm against collocated AERONET observations during January, February,**
820 **and March 2013 over China. a) Operational DT AOD, b) an intermediate AOD retrieved using the same LUT as the operational DT but with modified masking, c) AOD retrieved with the full regional research algorithm, and d) the extra collocations that are in c) but not in a).**

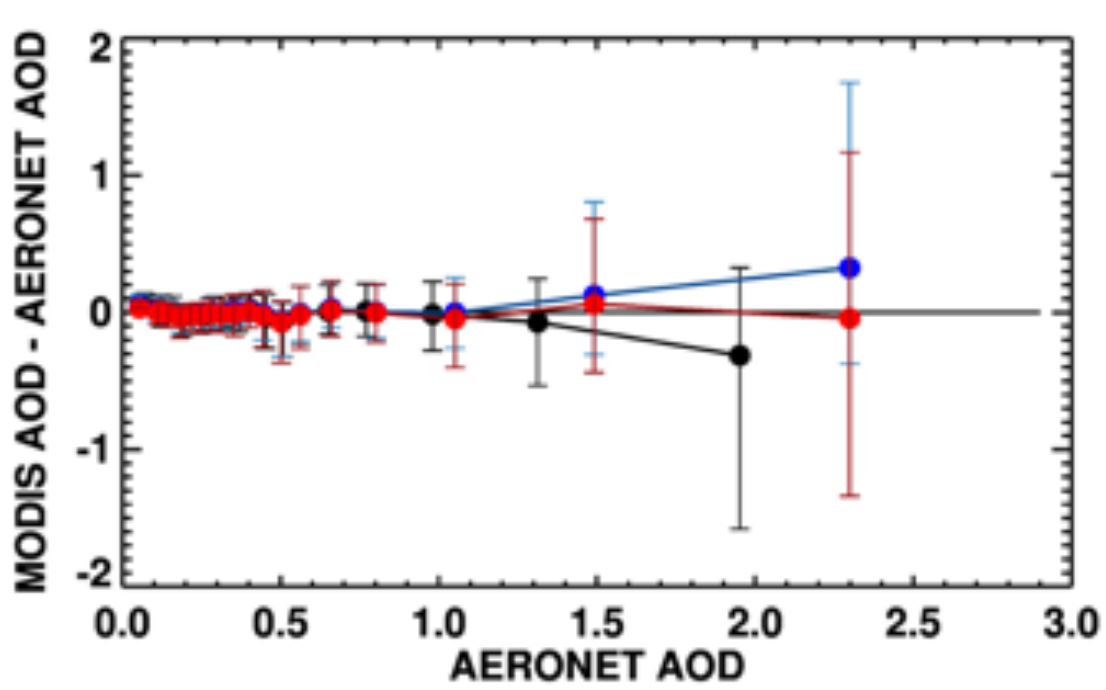

**Figure 10 Bias between MODIS and AERONET over land AOD at 0.55 µm as function of AERONET AOD at 0.55 µm. Black**
825 **represents the operational DT AOD, blue represents the AOD using the operational LUT but with new masks, and red represents**

**the research AOD. The dots are the mean bias within each AERONET AOD bin, and the bars represent the standard deviation of the bias.**

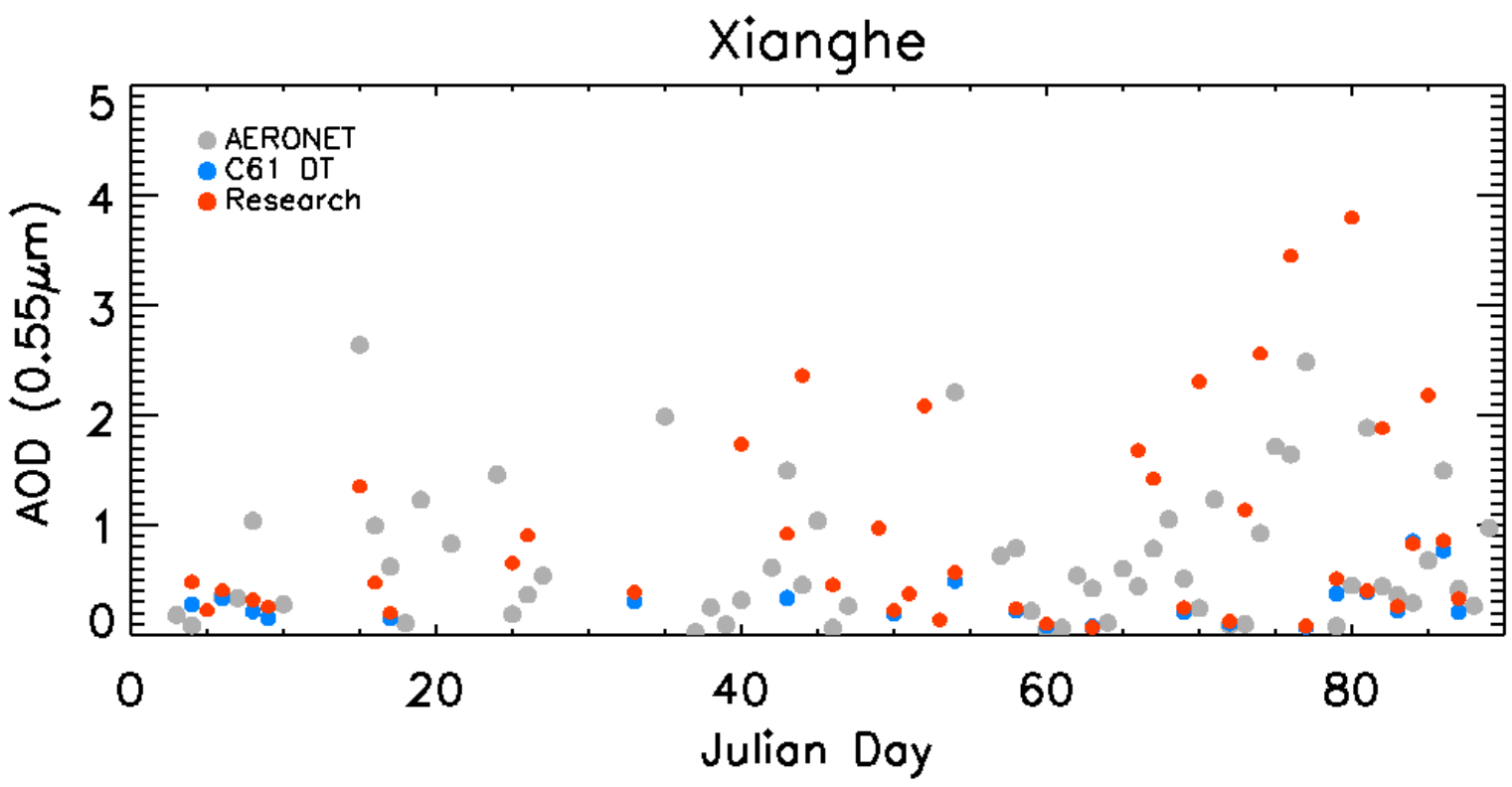

830 **Figure 11 Time series of daily averaged AERONET observations of AOD at 0.55 µm (in grey) as a function of Julian Day in 2013, and the corresponding daily operational MODIS DT (in blue) and research (in red) AOD over the Xianghe AERONET site.**

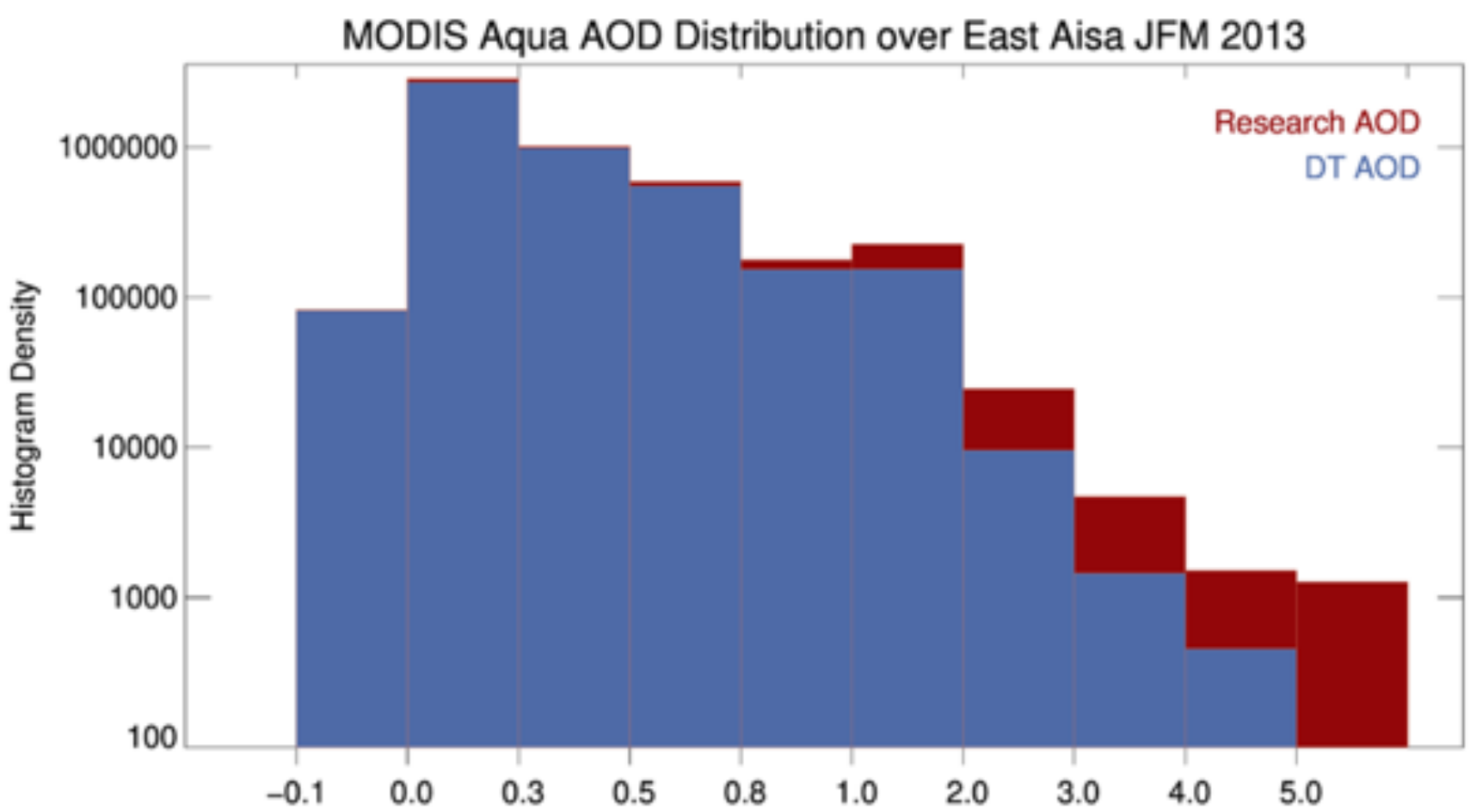

**Figure 12 The histogram of MODIS AOD over the study region from January to March 2013 in a logarithmic scale. The red is the research AOD, the blue is the operational AOD.**

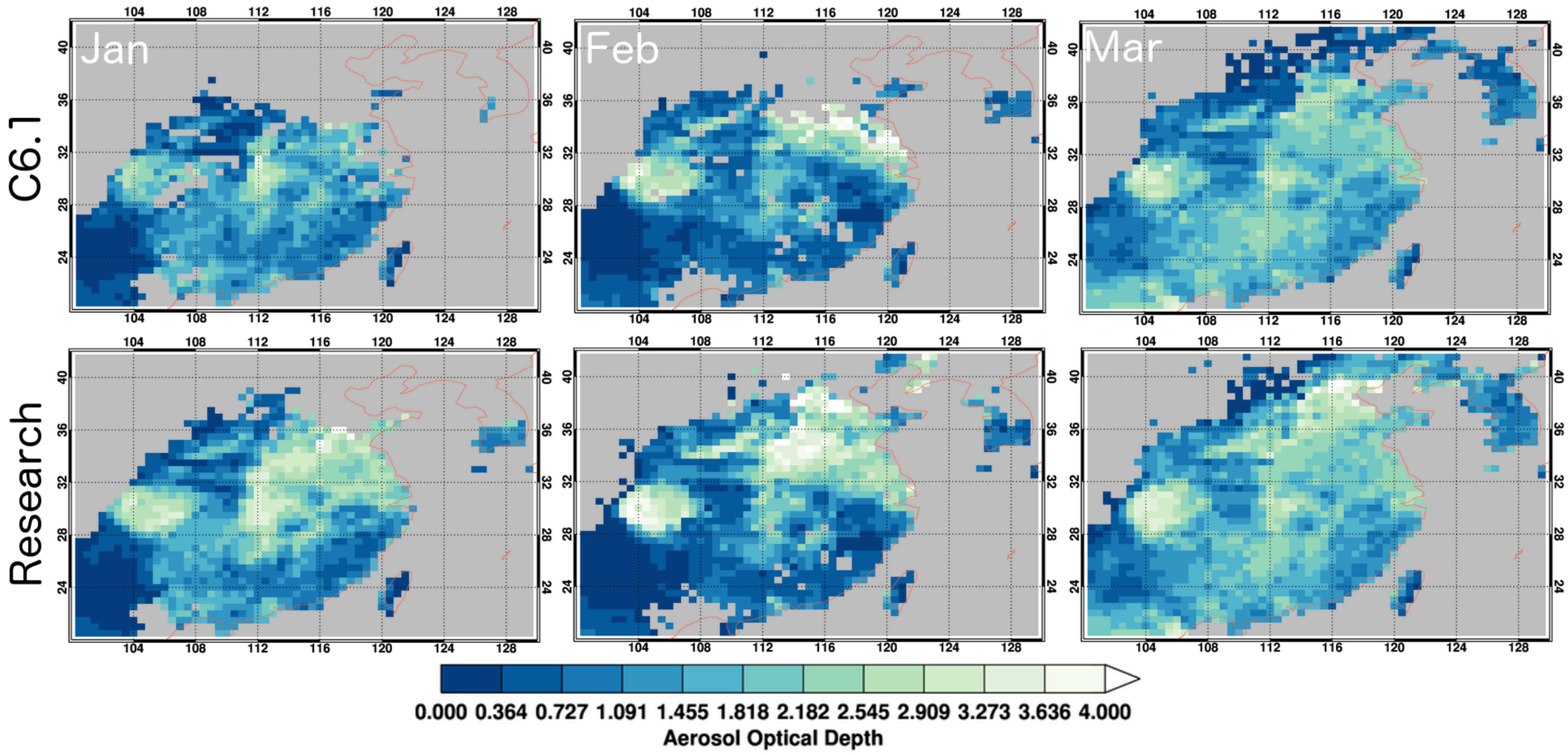

**Figure 13 Spatial distribution of averaged AOD from the operational product (upper row) and the research product (lower row) at 0.5° resolution over the study domain from January to March 2013.**

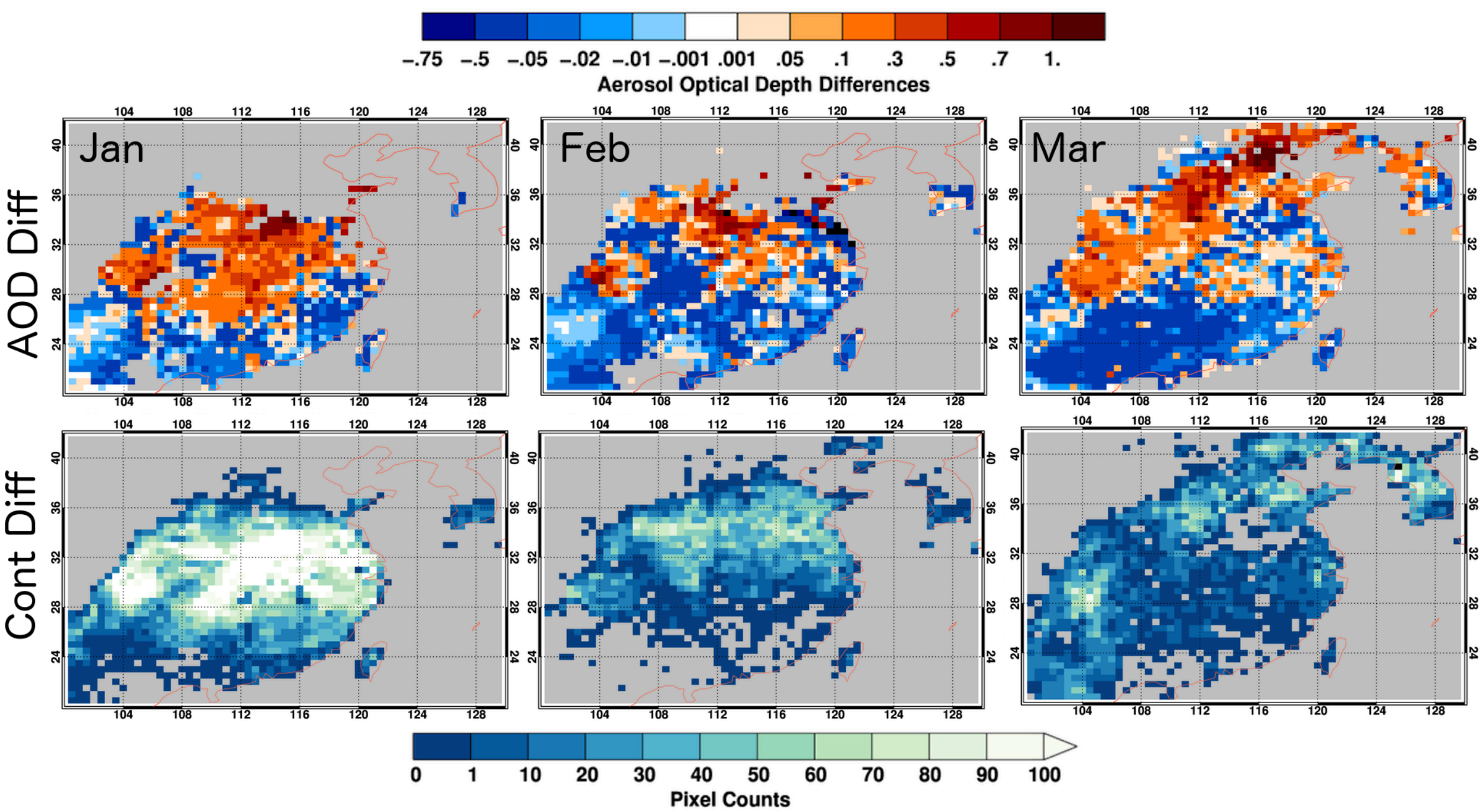

**Figure 14 Spatial distribution of AOD differences (upper row) and number of data points differences (lower row) between the operational product and the research product (research minus operational) at 0.5° resolution over the study domain from January to March 2013.**