# Peer review of "A Dark Target research aerosol algorithm for MODIS observations over eastern China: Increasing coverage while maintaining accuracy at high aerosol loading"

_Atmospheric Measurement Techniques, 2020_

## Referee Comment (RC1) · Anonymous Referee #1 · 14 Dec 2020

The authors changed the inland water and snow mask of original MODIS DT algorithm to increase data coverage in China. The aerosol model and the aerosol layer scale height are also changed to increase the accuracy of AOD inversion. The article has a complete structure and clear logic. However, I still have some major comments I hope the authors will explain before publishing as follows:

1. The inland water and snow mask threshold setting of MODIS is strict. Is it a strict mask to reduce the inaccuracy of inversion, or just a misjudgment? From the comparisons in Fig.9b, although the original DT algorithm can retrieve AOD from the additional

[Figure]

sample size from new mask, the accuracy has decreased. After using the new aerosol model and aerosol layer scale height (Fig.9c), there is not a significant improvement in accuracy, and the percentage of "within EE" also dropped. If the mask causes a decrease in the inversion accuracy, please comment on the impact of changing the mask conditions on the availability of AOD.

2. The author divided geographical areas to obtain three aerosol types, but does not rule out that aerosol type during the observation period find significant changes, such as dust weather. It is suggested that the author use cluster analysis or other methods to classify aerosol types, or discuss the frequency and contribution of special weather. Compared with the original MODIS model, how does the aerosol model proposed in this study contribute to AOD inversion?

3. In this research algorithm, the authors changed the aerosol layer scale height in the vertical profile in order to obtain better inversion results. However, the scale height is not always 0.5 km in all weather conditions. How did the authors choose the scale height under different weather conditions? If the scale height is always set to 0.5km, how much biases can be caused when retrieving the AOD from January to March 2013?

4. Figure 9 shows the validations of the research algorithm, but the advantages of the new algorithm cannot be clearly seen. It is recommended to show the advantages of the new algorithm point by point based on the results of Fig.9b and c like Fig.11. Similarly, the ordinate of Fig10 is too large to see the advantage of the new algorithm. Please adjust the ordinate to a reasonable range.

5. MODIS products have a resolution of at least 10km, and the research algorithm in this paper seems to be unlimited on the spatial resolution. So, why use $0.5°$ resolution for comparison in the 2013 winter characteristics analysis? Please explain.

---

## Short Comment (SC1) · 29 Dec 2020

This is a nice piece of work showcasing the use of MODIS data for aerosol applications. My question is related to the NDSI metric that relies on the reflectance from the 1.24 micron channel of MODIS. In the case of Terra MODIS, this channel has experienced significant degradation in its radiometric quality due to the electronic crosstalk and optical leak from the LWIR bands, especially after the Feb, 2016 safe-hold event. Recent investigations and subsequent improvements are expected to mitigate these artifacts in the next version of MODIS L1B (Collection 7). It is not entirely clear whether Terra

or Aqua MODIS is used for 2013 case as well as 2018 case. If Terra MODIS data from 1.24 micron was indeed used in this analysis, it would be interesting to see whether the radiometric artifacts mentioned above contribute towards misclassification of the pixels classified as snow, melting snow, or contaminated snow around the edges.

———————————————————

---

## Referee Comment (RC2) · Anonymous Referee #2 · 31 Dec 2020

I support the publishing of the paper after taking the following aspects into account.

I am a little bit surprised to see which papers are used to support the goal of this study, since the authors belong to the official product team, I believe they should be more careful of this issue. For instance, the authors mainly used Yan et al., 2016, Bilal et al., 2014, 2015, Wei et al., 2019, as the most important starting point of this work, the work of Yan et al., 2016 somehow fits the goal of this study, Wei et al., 2019 mainly raised that Dark-Target cannot provide aerosol product over the bright surfaces, which, to my understanding, does not fit to support why we need a new mask. The work of

Bilal et al., 2014, 2015 are using a single scattering assumption for the aerosol model to support the retrieval of haze conditions, if the authors cite these papers, does it mean the authors support the idea of using a single scattering assumption for aerosol retrieval? I will strongly suggest the authors re-check the published papers carefully.

Eq (1) and (4) should be re-format

Section 4.1, I believe that the paper from Yang et al., (2020) is some early test of this mask issue, however, the paper is in Chinese and I can only read the abstract part, how the heritage of new mask from MERSI has been adapted to MODIS, taking potential problems of instrument differences.

Section 4.2, I think it makes not much sense to compare the operational aerosol models, which to my understanding, were derived from other sites compared to the new regional model, which is specifically derived for China, the issues are how they define the region in which the "regional type" can be used? Or they define the whole of China, or the Eastern part, as regions with a "regional type"? A geographic figure to show regions assigned to "regional type" is helpful. Is the "regional type" a single-model or also a mixture of fine and coarse modes, if so, why there is only one mode presented in Table 1?

Is there any problem with Fig 9 (b), the correlation coefficient is so low? There is quite a large reduction of values fall into EE, can the author explain a bit more about this? We can also see that in Fig 9 (d), the performance of the additional points seems to be worse than the operational products, why?

Fig 12 should be updated, it is better to have some transparency for the overlap regions.

---

## Short Comment (SC2) · 6 Jan 2021

This work was interesting, which inspired me from another view to think about the strict cloud mask in the operational algorithm that maybe caused by inappropriate inland mask or snow mask. But some questions have been bothering me when I learned from this artical, please take several minitues to help me undertand clearly. Q1: Section 3, you talked about case 1 (high AOD pollution) and case 2 (a low pollution), which means that you wanted to improve the operational mask from these two points, but in the Section 5 and Section 6, you spent a lot of time describing the validation results

of case 1, it will be a complete story if you focus on the better performance of case 2. For example, Figure 10 clearly indicates a better improvement in high AOD values of research algorithm, but when OAD falls within the range of 0∼1.0, the differences of the three products almost overlap, and it is difficult to distinguish which is better.

Q2: As presented in Table 1, it is difficult to conclude the 'Also notice that the moderate absorbing aerosol model shows increased absorption with increasing AOD, which is opposite to the non-abosrbing model as well as to the regional model.' And 'The differences in the imaginary part of the refractive index show that when compared with . . . . . .ïïjǏ when AOD> ∼2, . . . . . . when AOD<0.5. . . . . .', you have not shown any figure or table to prove this. As for the real part of refractive index, the regional type has more strong scattering effect.

―――――――――――――――――――

---

## Short Comment (SC3) · 12 Jan 2021

Title: A Dark Target research aerosol algorithm for MODIS observations over eastern China: Increasing coverage while maintaining accuracy at high aerosol loading

Authors: Yingxi R. Shi, Robert C. Levy, Leiku Yang, Lorraine A. Remer, Shana Mattoo, and Oleg Dubovik

I have a few comments below that are specific to the cases that were identified as high AOD pollution days by the research algorithm but were not identified as AOD>1 by

[Figure]

AERONET.

Abstract: "We also find that the research algorithm is able to identify additional pollution events that a triad of AERONET instruments surrounding Beijing could not." First, related to the triad of AERONET sites please note somewhere in the text that the Beijing-CAMS site was missing data from February 9 through March 12 (over one month) due to an equipment issue.

Lines 460-465: "There were 17 days when the research product identified a polluted day but AERONET did not, and 7 days when AERONET observed AOD > 1.0 but the research algorithm did not capture the event. It is easy to understand when AERONET identified a polluted day but the research retrieval did not, because the AERONET observation time can be different from MODIS overpass time. The polluted scene can be cloud covered at over pass, but be captured by AERONET before or after, or the scene can significantly change between two observing times. It is more difficult to understand how the research algorithm could identify a pollution event on 17 days that all three AERONET stations missed."

I also was curious why AERONET would miss 17 high pollution days that were identified by the new MODIS research algorithm. When I looked at the AERONET data and MODIS images for all 17 of these days "missed by AERONET" the reasons became clearer to me. In my opinion these days fell into four general categories: (1) Cloudy days in MODIS images (both Terra and Aqua) with a lack of AERONET data therefore these seem to me to be likely misidentification of clouds as high AOD pollution by the research algorithm. (2) Days with little or no cloud cover but and with much AOD data from AERONET in Level 2 (V3). However the AOD as measured by AERONET on these days was <1 at 550 nm, sometimes by 0.10-0.50 lower so this falls within the scatter of the research algorithm AOD versus AERONET measurements in Figure 9c of the paper. Spatial variance of AOD can explain some of this scatter so this is not all satellite algorithm uncertainty (see related category 4 below). (3) Days where there was no Level 2 AERONET AOD with AOD(550)>1, however there was L1 data

with AOD(550)>1 that the SDA algorithm identified as fine mode, therefore pollution. Eck et al. (2018) found that for the Xianghe site there were 15% of high AOD days (AOD(500)>1) in that were screened from L2 in V3 but had fine mode AOD(500)>1 in L1 data. Therefore, AERONET did detect these pollution events but the cloud screening and/or V3 QC eliminated them. This reference could be used to help explain these cases. I also include in this category days with only a few L1 data and the shortest wavelength of AOD measured by AERONET was >440 nm yet Angstrom Exponent was moderate (∼1), since the limits of sun photometry prevented the measurement of the full wavelength range AOD spectra (nearly complete attenuation of shorter wavelength direct sun signal). (4) Days where there was on obvious gradient of AOD in the Beijing region from the MODIS images, therefore the higher AOD from the research algorithm could very likely come from haze that was in the region but not located over Beijing therefore the AERONET sensors could not detect it.

Days 'missed by AERONET' but identified by the research algorithm in each category (note that some days have characteristics of multiple categories): Category (1): 13, 36, 41, 45, 49, 55, 61, 78 Category (2): 40, 51, 54, 57, 58, 64, 82, Category (3): 12, 13, 55, 82, 86 Category (4): 51, 54, 58, 78, 82, 86 Cat (1) = Extensive cloud cover Cat (2) = AERONET AOD measured but < 1 at 550 nm Cat (3) = AERONET L1 data with AOD(550)>1 but no L2 data at high AOD Cat (4) = Gradient in AOD with lower AOD over AERONET sites

Obviously, I do not think that it is accurate to label these 17 days as pollution events that were 'missed by AERONET'. I suggest that you should include some of the issues I have identified above in the discussions in your paper as they may help explain some of these discrepancies between AERONET measurements and MODIS retrievals of AOD, even if you disagree somewhat with some of my categorizations.

I have one other unrelated comment. This study is a valuable seasonal investigation of high AOD events in the area around Beijing for the months of January through March 2013. The research algorithm shows significant improvement over the operational

one for high AOD events, in part due to improved earth surface characteristics classification. The AOD in June through August is much higher in the Beijing region than in winter (on average ∼50% higher), and this high AOD is often associated with significant cumulus cloud cover. It would be useful to also test the research algorithm in this same region in summer when surface effects would be less important but cloud effects (humidification and cloud processing) and cloud screening are more dominant issues in satellite retrievals. Perhaps in a follow-on study?

Please also note the supplement to this comment:
https://amt.copernicus.org/preprints/amt-2020-450/amt-2020-450-SC3-supplement.pdf

---

## Author Comment (AC1) · 12 Mar 2021

Reply to referee #1

The authors changed the inland water and snow mask of original MODIS DT algorithm to increase data coverage in China. The aerosol model and the aerosol layer scale height are also changed to increase the accuracy of AOD inversion. The article has a complete structure and clear logic. However, I still have some major comments I hope the authors will explain before publishing as follows:

We thank reviewer for his/her suggestions. Below is our point to point response.

1. The inland water and snow mask threshold setting of MODIS is strict. Is it a strict mask to reduce the inaccuracy of inversion, or just a misjudgment? From the comparisons in Fig.9b, although the original DT algorithm can retrieve AOD from the additional sample size from new mask, the accuracy has decreased. After using the new aerosol model and aerosol layer scale height (Fig.9c), there is not a significant improvement in accuracy, and the percentage of "within EE" also dropped. If the mask causes a decrease in the inversion accuracy, please comment on the impact of changing the mask conditions on the availability of AOD.

The reviewer has identified the esoteric problem with aerosol remote sensing with a sensor like MODIS or VIIRS. The multi-spectral measurements contain sufficient information to accurately retrieve some aerosol characteristics, such as AOD, if the other properties involved in the retrieval, such as particle size, absorption and surface reflectance, can be sufficiently constrained. The algorithms must resort to a pre-computed Look Up Table (LUT) and empirical constraints on surface reflectance. The standard global Dark Target algorithm has fine-tuned its assumptions to constrain its retrieval accuracy to well-defined error bars, in a global sense. However, part of the algorithm's success is based on a careful masking of situations that will not match assumptions. In some situations, such as the China-in-winter example that we focus on here, this masking results in the unfortunate loss of a significant number of retrievals with high aerosol loading. This biases the overall statistics of the resulting AOD and reduces product availability for applications that require day-to-day AOD monitoring.

Thus, the main purpose of this exercise reported on in this paper is to increase the number of AOD retrievals over China. We know a priori that to bring back the once-masked high AOD will immediately introduce a degradation of overall accuracy, because that is the reason these opportunities were rejected in the first place. We also know that adjusting the LUT and other assumptions to improve the quality of the new retrievals will likely shift old retrievals to poorer accuracy. Thus a priori goals are (1) to bring back high AOD, (2) make adjustments to reduce **new biases** introduced by the new retrievals and (3) to minimize new error and scatter introduced across the range of AOD.

There are trade-offs in trying to meet all three goals. We proceeded with this trade-off by making increased number of retrievals (goal 1) and managing new biases (goal 2).

The new algorithm shows significant increase of number of retrievals along with reduced bias when aerosol loading is high.  The result is a success in what we set out to do, but unfortunately at the expense of increasing the scatter across the range of AOD.

To make the success criteria clearer, we have modified the text (lines 68-71) and Figures 9 and 10 in the revision and have also added Table 2.

2. The author divided geographical areas to obtain three aerosol types but does not rule out that aerosol type during the observation period find significant changes, such as dust weather. It is suggested that the author use cluster analysis or other methods to classify aerosol types or discuss the frequency and contribution of special weather. Compared with the original MODIS model, how does the aerosol model proposed in this study contribute to AOD inversion?

The focus of this study is that DT loses data coverage over winter.  Thus, we focus on January to March when spring dust storms are rare. Nevertheless, we generated the aerosol model using AERONET's size distribution which include both fine and coarse particles and represent the averaged aerosol properties, as measured by AERONET, over the region we selected during the study months.  The average AERONET models for each of the geographical areas turned out to be sufficiently similar so that we can use the overall mean and not differentiate from region to region. Cluster analysis would tell us the same thing.

We realize now that the way we presented this work leads to the wrong expectations for a reader. In the revision we approach the description of finding a regional model differently, explaining right up front the conclusion that there is insufficient evidence to use more than one aerosol model for all of China.  Then the plots of AERONET-derived size distribution from the different parts of China are put into better context.

The regional model has a stronger AOD dependency in terms of absorption when compared with non-absorbing and moderate models from the standard retrieval. It is slightly more absorbing in low AOD compared to the standard non-absorbing model (and similar to the moderate model) and its absorption decreases with increase of AOD. The regional model becomes less absorbing than both standard models when AOD > 2. Thus, the new aerosol model results in a slight increase of AOD when AOD < 2 and lower AOD when AOD > 2.

3. In this research algorithm, the authors changed the aerosol layer scale height in the vertical profile in order to obtain better inversion results. However, the scale height is not always 0.5 km in all weather conditions. How did the authors choose the scale height under different weather conditions? If the scale height is always set to 0.5km, how much biases can be caused when retrieving the AOD from January to March 2013?

We agree that the best approach is to change aerosol layer height for each retrieval scene. However, there is no information for us to get the prior knowledge of aerosol layer height nor does the current DT algorithm structure support a selection of aerosol layer height for each retrieval. Our analyses show that when the scale height is set to 0.5 km, the low bias when AOD < 0.5 (as seen in Fig. 9 and Figure 10.) is small. Analyses regarding this issue is also discussed in the reply of next question.

4. Figure 9 shows the validations of the research algorithm, but the advantages of the new algorithm cannot be clearly seen. It is recommended to show the advantages of the new algorithm point by point based on the results of Fig.9b and c like Fig.11. Similarly, the ordinate of Fig10 is too large to see the advantage of the new algorithm. Please adjust the ordinate to a reasonable range.

Thanks for the suggestion, we modified Figure 9 and Figure 10 and added a Table to show statistics.

Figure 9 shows how three AOD data sets, namely operational DT AOD, intermediate AOD retrieved using the same LUT as the operational DT but with modified masking (New Mask), and AOD from the research algorithm, compare to each other. All statistics are shown in Table 2. Figure 9a overlays the operational DT AOD onto the New Mask AOD. We can easily identify paired data from the two datasets. The slight differences between two paired data points are expected because these data points represent spatially averaged MODIS AOD and temporally averaged AERONET AOD. When within the averaging criteria new MODIS AOD become available in the New Mask AOD, the averaged value will be slightly different.

Figure 9a shows there is a large (50%) increment in **the number of retrievals** in the New Mask AOD when AERONET AOD > 1 (19 points from operational DT and 30 from New Mask). **This is the primary goal, as explained in 1. above**. These additional points show that our algorithm successfully retrieved AOD from many high aerosol scenes that are not retrieved in the operational algorithm. However, these extra AOD are highly overestimated with a mean bias of 0.26 when AOD > 1 while the Operational DT shows a negative mean bias of -0.196. Figure 9a also shows multiple New Mask AOD exist without corresponding Operational AOD when AOD is around 0.8 (about 20% increment in number of AOD < 1), which are most likely due to change of snow mask. Figure 9b overlays the research DT AOD on top of the New Mask AOD. We notice that there are large reductions of AOD values when New Mask AOD are above 1.5. The mean bias of the entire data set reduced from 0.16 in New Mask AOD (0.26 from AOD > 1) to 0.076 in the Research AOD (0.097 from AOD > 1). **Reducing bias is the second goal from 1. above**.
The RMSE also reduced from 0.517 to 0.45, although it is still larger than in the operational DT algorithm. This is the tradeoff we are forced to live with.

These are significant reductions in bias after applying our new aerosol model with a much stronger AOD dependent absorption and using a reduced aerosol layer height. We can also see from Figure 9b that when AOD is lower than 0.5, there are no obvious low points from Research AOD when compared with New Mask AOD, meaning that the change in aerosol model and aerosol layer height has minimum effects when AOD is low. Thus, although we are forced to use one aerosol layer height in the retrieval process that is representative of heavy aerosol loading conditions, the impacts of this choice are small on AOD retrievals when aerosol loading is low. A similar conclusion is also shown in Figure 10. We changed the y-axis data range in Figure 10 to better illustrate data when AERONET AOD is small. We can see that when AERONET AOD is less than 0.5, the mean error pattern and standard deviation of the bins from three data sets are closely following each other. But they diverge at AOD > 1.

[Figure]

Figure 9 Comparisons of the MODIS DT AOD at 0.55 μm against collocated AERONET observations during January, February, and March 2013 over China. Three datasets are used, operational DT AOD (Operational DT), an intermediate AOD retrieved using the same LUT as the operational DT but with modified masking (New Mask), and AOD retrieved with the full regional research algorithm (Research). a) Operational DT AOD overlay on New Mask AOD, b) Research AOD overlay on New Mask AOD.

[Figure]

Figure 10 Bias between MODIS and AERONET over land AOD at 0.55 µm as function of AERONET AOD at 0.55 µm. Black represents the operational DT AOD, blue represents the AOD using the operational LUT but with new masks (New Mask), and red represents the research AOD. The dots are the mean bias within each AERONET AOD bin, and the bars represent the standard deviation of the bias.

Table 2 Statistics of validation between Operational DT AOD, AOD using the operational LUT but with new masks (New Mask), and Research AOD against AERONET during January, February, and March 2013 over China. Numbers in parentheses are the statistics for AERONET AOD > 1.

|  | % within EE | N | $R^2$ | Mean Bias | RMSE | Slope | Offset |
|---|---|---|---|---|---|---|---|
| Operational DT | 40.91 | 66(19) | 0.754 | 0.003 (-0.196) | 0.286 | 0.75 | 0.151 |
| New Mask DT | 30.34 | 88(28) | 0.700 | 0.161 (0.260) | 0.517 | 1.01 | 0.098 |
| Research DT | 33.71 | 89(30) | 0.701 | 0.076 (0.097) | 0.450 | 0.96 | 0.081 |

5. MODIS products have a resolution of at least 10km, and the research algorithm in this paper seems to be unlimited on the spatial resolution. So, why use 0.5 resolution for comparison in the 2013 winter characteristics analysis? Please explain.

The gridded product represents the mean states of the aerosol loading over a region and within a time window, while pixel level data show variations of the aerosols loading at certain spot over a period of time. In this section, we want to investigate the change in aerosol spatial distribution due to increasing high AOD retrievals over winter. It shows the bulk impact of research products. Thus, we want to use gridded data and 0.5 by 0.5 degree grid box to insure enough data points in each grid. The impact on pixel level data is shown in the AOD histogram (Fig. 12).

---

## Author Comment (AC2) · 12 Mar 2021

Reply to referee #2

I support the publishing of the paper after taking the following aspects into account. I am a little bit surprised to see which papers are used to support the goal of this study, since the authors belong to the official product team, I believe they should be more careful of this issue. For instance, the authors mainly used Yan et al., 2016, Bilal et al., 2014, 2015, Wei et al., 2019, as the most important starting point of this work, the work of Yan et al., 2016 somehow fits the goal of this study, Wei et al., 2019 mainly raised that Dark-Target cannot provide aerosol product over the bright surfaces, which, to my understanding, does not fit to support why we need a new mask. The work of Bilal et al., 2014, 2015 are using a single scattering assumption for the aerosol model to support the retrieval of haze conditions, if the authors cite these papers, does it mean the authors support the idea of using a single scattering assumption for aerosol retrieval? I will strongly suggest the authors re-check the published papers carefully.

We thank the reviewer to encourage us to perform a more thorough background study.  We cite these paper as they all mention the problem of DT missing retrievals over eastern China and tried different methods to solve the issue.  It doesn't mean that we support these solutions because in this study we developed a new solution that can be applied to operational production in near real time. We now state this explicitly, and we added following citations:

Li S, Chen L, Xiong X, Tao J, Su L, Han D, Liu Y. Retrieval of the haze optical thickness in North China Plain using MODIS data. IEEE transactions on geoscience and remote sensing. 2012 Oct 10;51(5):2528-40. DOI: 10.1109/TGRS.2012.2214038

Zhang X, Wang H, Che HZ, Tan SC, Shi GY, Yao XP, Zhao HJ. Improvement of snow/haze confusion data gaps in MODIS Dark Target aerosol retrievals in East China. Atmospheric Research. 2020 May 30:105063. DOI: 10.1016/j.atmosres.2020.105063.

Chen W, Fan A, Yan L. Performance of MODIS C6 aerosol product during frequent haze-fog events: A case study of Beijing. Remote Sensing. 2017 May;9(5):496. DOI: 10.3390/rs9050496

Eq (1) and (4) should be re-format

We added parentheses and reformat the equations.

Section 4.1, I believe that the paper from Yang et al., (2020) is some early test of this mask issue, however, the paper is in Chinese and I can only read the abstract part, how the heritage of new mask from MERSI has been adapted to MODIS, taking potential problems of instrument differences.

MODIS and MERSI have very similar design and the two used channels are very similar: 0.645, 0.856, and 2.11 $\mu$m in MODIS and 0.654, 0.869, and 2.13 $\mu$m in MERSI.  Thus, after testing, we adapted the threshold used in MERSI ($\rho 2.11 < 0.08$).

Section 4.2, I think it makes not much sense to compare the operational aerosol models, which to my understanding, were derived from other sites compared to the new regional model, which is specifically derived for China, the issues are how they define the region in which the "regional type" can be used? Or they define the whole of China, or the Eastern part, as regions with a "regional type"? A geographic figure to show regions assigned to "regional type" is helpful. Is the "regional type" a single-model or also a mixture of fine and coarse modes, if so, why there is only one mode presented in Table 1?

We realize now that the way we presented this work leads to the wrong expectations for a reader. In the revision we approach the description of finding a regional model differently, explaining right up front the conclusion that there is insufficient evidence to use more than one aerosol model for all of China. Then the plots of AERONET-derived size distribution from the different parts of China are put into better context.

We applied a single aerosol model to the entire study area that is shown in Figure 13 and 14, calling it our "regional" model, as opposed to the more general global models used by the operational algorithm. From Figure 4, we see that subareas 1, 2, and 3 are relatively similar especially for AOD bins < 2, and AOD > 3 in Area 3 is represented by only one data. Thus, we aggregated aerosol models from all three subareas into one and applied to the entire study region. The regional model is bi-model distribution that contains both fine and coarse mode as what Figure 5 shows. However, although including a coarse mode portion, this aerosol model is still fine mode dominate and is listed as fine mode model shown in Table 1. During the study period, the dominate aerosol type is pollution. Thus, we didn't change the coarse mode aerosol model, which is made from dust retrievals by AERONET.

Is there any problem with Fig 9 (b), the correlation coefficient is so low? There is quite a large reduction of values fall into EE, can the author explain a bit more about this?
We can also see that in Fig 9 (d), the performance of the additional points seems to be worse than the operational products, why?

Thanks for pointing this out, we found that there are two outliers, which cause this issue. We removed these points from our analyses due to these two outlier values are way above 5. These two points values reduced largely after adapting the new LUT and values are within 5.

We will reiterate here what we wrote to Reviewer #1 who raised similar concerns.

The reviewer has identified the esoteric problem with aerosol remote sensing with a sensor like MODIS or VIIRS. The multi-spectral measurements contain sufficient information to accurately retrieve some aerosol characteristics, such as AOD, if the other properties involved in the retrieval, such as particle size, absorption and surface reflectance, can be sufficiently constrained. The algorithms must resort to a pre-computed Look Up Table (LUT) and empirical constraints on surface reflectance. The standard global Dark Target algorithm has fine-tuned its assumptions to constrain its retrieval accuracy to well-defined error bars, in a global sense. However, part of the algorithm's success is based on a careful masking of situations that will

not match assumptions. In some situations, such as the China-in-winter example that we focus on here, this masking results in the unfortunate loss of a significant number of retrievals with high aerosol loading. This biases the overall statistics of the resulting AOD and reduces product availability for applications that require day-to-day AOD monitoring.

Thus, the main purpose of this exercise reported on in this paper is to increase the number of AOD retrievals over China. We know a priori that to bring back the once-masked high AOD will immediately introduce a degradation of overall accuracy, because that is the reason these opportunities were rejected in the first place. We also know that adjusting the LUT and other assumptions to improve the quality of the new retrievals will likely shift old retrievals to poorer accuracy. Thus a priori goals are (1) to bring back high AOD, (2) make adjustments to reduce new biases introduced by the new retrievals and (3) to minimize new error and scatter introduced across the range of AOD.

There are trade-offs in trying to meet all three goals. We proceeded with this trade-off by making increased number of retrievals (goal 1) and managing new biases (goal 2). The new algorithm shows significant increase of number of retrievals along with reduced bias when aerosol loading is high. The result is a success in what we set out to do, but unfortunately at the expense of increasing the scatter across the range of AOD.

To make the success criteria clearer, we have modified the text (lines 68-71) and Figures 9 and 10 in the revision and have also added Table 2.

Figure 9 shows how three AOD data sets, namely operational DT AOD, intermediate AOD retrieved using the same LUT as the operational DT but with modified masking (New Mask), and AOD from the research algorithm, compare to each other. All statistics are shown in Table 2. Figure 9a overlays the operational DT AOD onto the New Mask AOD. We can easily identify paired data from the two datasets. The slight differences between two paired data points are expected because these data points represent spatially averaged MODIS AOD and temporally averaged AERONET AOD. When within the averaging criteria new MODIS AOD become available in the New Mask AOD, the averaged value will be slightly different.

Figure 9a shows there is a large (50%) increment in the number of retrievals in the New Mask AOD when AERONET AOD > 1 (19 points from operational DT and 30 from New Mask). This is the primary goal. These additional points show that our algorithm successfully retrieved AOD from many high aerosol scenes that are not retrieved in the operational algorithm. However, these extra AOD are highly overestimated with a mean bias of 0.26 when AOD > 1 while the Operational DT shows a negative mean bias of -0.196. Figure 9a also shows multiple New Mask AOD exist without corresponding Operational AOD when AOD is around 0.8 (about 20% increment in number of AOD < 1), which are most likely due to change of snow mask. Figure 9b overlays the research DT AOD on top of the New Mask AOD. We notice that there are large reductions of AOD values when New Mask AOD are above 1.5. The mean bias of the entire data set reduced from 0.16 in New Mask AOD (0.26 from AOD > 1) to 0.076 in the Research AOD (0.097 from AOD > 1). Reducing bias is the second goal.

The RMSE also reduced from 0.517 to 0.45, although it is still larger than in the operational DT algorithm. This is the tradeoff we are forced to live with.

These are significant reductions in bias after applying our new aerosol model with a much stronger AOD dependent absorption and using a reduced aerosol layer height. We can also see from Figure 9b that when AOD is lower than 0.5, there are no obvious low points from Research AOD when compared with New Mask AOD, meaning that the change in aerosol model and aerosol layer height has minimum effects when AOD is low. Thus, although we are forced to use one aerosol layer height in the retrieval process that is representative of heavy aerosol loading conditions, the impacts of this choice are small on AOD retrievals when aerosol loading is low. A similar conclusion is also shown in Figure 10. We changed the y-axis data range in Figure 10 to better illustrate data when AERONET AOD is small. We can see that when AERONET AOD is less than 0.5, the mean error pattern and standard deviation of the bins from three data sets are closely following each other. But they diverge at AOD > 1.

[Figure]

Figure 9 Comparisons of the MODIS DT AOD at 0.55 μm against collocated AERONET observations during January, February, and March 2013 over China. Three datasets are used operational DT AOD (Operational DT), an intermediate AOD retrieved using the same LUT as the operational DT but with modified masking (New Mask), and AOD retrieved with the full regional research algorithm (Research). a) Operational DT AOD overlay on New Mask AOD, b) Research AOD overlay on New Mask AOD.

[Figure]

Figure 10 Bias between MODIS and AERONET over land AOD at 0.55 μm as function of AERONET AOD at 0.55 μm. Black represents the operational DT AOD, blue represents the AOD using the operational LUT but with new masks (New Mask), and red represents the research AOD. The dots are the mean bias within each AERONET AOD bin, and the bars represent the standard deviation of the bias.

Table 2 Statistics of validation between Operational DT AOD, AOD using the operational LUT but with new masks (New Mask), and Research AOD against AERONET during January, February, and March 2013 over China. Numbers in parentheses are the statistics for AERONET AOD > 1.

|  | % within EE | N | $R^2$ | Mean Bias | RMSE | Slope | Offset |
|---|---|---|---|---|---|---|---|
| Operational DT | 40.91 | 66(19) | 0.754 | 0.003 (-0.196) | 0.286 | 0.75 | 0.151 |
| New Mask DT | 30.34 | 88(28) | 0.700 | 0.161 (0.260) | 0.517 | 1.01 | 0.098 |
| Research DT | 33.71 | 89(30) | 0.701 | 0.076 (0.097) | 0.450 | 0.96 | 0.081 |

Fig 12 should be updated, it is better to have some transparency for the overlap regions.

The reason for adding transparency is to see both bars when one is higher than another. However, after we updated Figure 12 regarding the smaller domain over China (25° to 40° N and 105° to 120° E). The updated figure shows that in all bins, red bars are higher than blue bars. Thus, we do not use transparency plot due to it is unnecessary in this figure.

---

## Author Comment (AC3) · 12 Mar 2021

Example figure, A pollution event over eastern China on Terra MODIS sensor with granule id of 2013348.0310. a) MODIS Terra RGB image, b) MODIS Dark Target (DT) AOD at .0.55 μm for all available retrievals (QA = 0 to 3), c) Research AOD at 0.55 μm, for all available retrievals, d) Snow mask, e) 11 μm Brightness Temperature map. The red line shows the cloud edge.

---

## Author Comment (AC4) · 12 Mar 2021

We thank Dr. Si's comment and are happy that our work is inspiring to the community. Here are our answers to your questions:

A1: We agree that both scenarios (high and low aerosol loading) should be evaluated. The main goal of our study is to increase the data coverage over eastern China. We expect that in achieving this goal, we may need to accept an overall degradation of accuracy. When the data coverage increases, the change of the data accuracy strongly

depends on the quality of the retrievals at high aerosol loading, which why the higher AOD ranges becomes the focus of the paper. Thus, it is important to re-assess our aerosol model assumption and to evaluate the uncertainties. To better illustrate the data performances against AERONET, we regenerated the Figure 9 and changed the y-axis range in Figure 10. The new figures include all ranges of AOD including the low AOD cases. The figure shows that when AOD < 0.5, even there is about a 30% increase in the number of retrievals, although there is indeed an overall decrease in accuracy, as expected. We do note that the overall bias in research AOD does not change much compared with the operational AOD. Please read our responses to Reviewers #1 and #2 for a more complete explanation of the study's goals and successes.

A2: In our Table 1 we provide equations of how imaginary part of the refractive index is calculated. The imaginary part of the refractive index related to absorption and are all as functions of AOD. The non-absorbing and regional model the sign in front of the term that includes AOD is positive, indicating that with increasing of AOD, the imaginary part is increasing. This sign is opposite in moderate absorbing model. Note that the change of absorption in terms of AOD is not linear as we seen in Figure 6, however, to avoid over fitting, we use linear relations which will have a better match at high AOD end. This choice was made because the assumption of absorption has much larger influence on retrieval uncertainty when AOD is large.

---

## Author Comment (AC5) · 12 Mar 2021

We thank Dr. Eck to provide detailed analyses on research algorithm selected heavy pollution days. With these analyses, we have a better understanding of the discrepancies between the two datasets. We also included descriptions of these analyses into text. The detailed changes are listed in the end. Among the four categories you defined, there are two scenarios: 1) we do not know whether or not heavy pollution exist due to cloud coverage. 2) There were pollutions existing and research algorithm and AERONET are not agreeing with each other. We agree with all the analyses on

scenario 2 and make a deeper look into scenario 1. When we categorize these days originally, the standard we used is whether or not heavy pollution exist. We used RGB images as well as DT/DB/MAIAC/VIIRS retrievals to facilitate our decision. We agree that cloud contamination may occur within the granule, but the identification of event is still valid. We also agree that if there is too little data to base on, the event should not be counted. Thus, we changed the number of research algorithm misclassified heavy polluted events from 3 to 5 (36, 41, 45, 49, 61). The description of the table is changed to following:

"It is more difficult to understand how the research algorithm could identify a pollution event on 17 days when all three AERONET stations do not report AOD > 1 at their quality-assured level (level 2). To begin we note that one of the three AERONET stations (Beijing-CAM5) was down for maintenance for more than a month during this time (T. Eck Short Comment in Interactive Discussion). Then, to confirm polluted days that the satellite identified but the operating AERONET stations did not, we visually compared each day using RGB images and MODIS DB and MAIAC AOD retrievals, as well as nearby over ocean AOD retrievals as a reference. Among these 17 days, 12 days have pollution present visually (with retrieval over cloud free/snow free land or ocean). Within these 12 days, analyses show two different scenarios lead to the discrepancies between AERONET and the research AOD. Scenario 1 includes the majority of the 12 days. In these days, AERONET Level 2 (V3) report AOD at 0.55 micron < 1, (0.50 to 0.90). Possible reasons for the differences can be (1) sampling differences, especially when an obvious gradient of AOD exists or (2) the uncertainty within the research product (see Figure 9). Scenario 2 consists of five days. These are days where there was no Level 2 AERONET AOD with AOD550 > 1, however there were L1 data. Eck et al. (2018) found that for the Xianghe site 15% of high AOD days (AOD500 > 1) never made it from L1 to L2. The 5 days identified by the satellite as pollution events but could not be confirmed by visual inspection were overcast with clouds (day 36, 41, 45, 49, and 61). In these five cases we expect cloud effects in the MODIS product that do not appear in the AERONET data are causing the AOD to exceed the AOD = 1.0 threshold.

We note that none of the five days in question have AOD over visually identified snow patches. Overall, we are happy with the ability of using the DT research product to identify pollution events, which can complement sparse ground observations."

The issue you mentioned on reducing data coverage due to cloud masking during summer time is also very interesting, we will continue look into this issue on follow-on study.
* * *

---

## Author Response (AR2)

Dear Dr. Leeuw,

Thank you very much for your comments. We modified all descriptions regarding Figure 9 in Section 5 following your list and throughout the abstract and conclusion. To answer your last question, we mentioned in our last couple sentences that we are trying to put this change to global scale, which then can be put into our operational code. Some tests are done, but more global testing is needed. The research data from this paper is definitely open for interested party to use.  Research data over this region beyond winter 2013 can be generated upon request as well.